# Switchable chiral transport in charge-ordered kagome metal CsV$_3$Sb$_5$

Chunyu Guo[1,2 ✉], Carsten Putzke[2], Sofia Konyzheva[1], Xiangwei Huang[1], Martin Gutierrez-Amigo[3,4], Ion Errea[3,5,6], Dong Chen[7,8], Maia G. Vergniory[5,7], Claudia Felser[7], Mark H. Fischer[9 ✉], Titus Neupert[9 ✉] & Philip J. W. Moll[1,2 ✉]

When electric conductors differ from their mirror image, unusual chiral transport coefficients appear that are forbidden in achiral metals, such as a non-linear electric response known as electronic magnetochiral anisotropy (eMChA)[1–6]. Although chiral transport signatures are allowed by symmetry in many conductors without a centre of inversion, they reach appreciable levels only in rare cases in which an exceptionally strong chiral coupling to the itinerant electrons is present. So far, observations of chiral transport have been limited to materials in which the atomic positions strongly break mirror symmetries. Here, we report chiral transport in the centrosymmetric layered kagome metal CsV$_3$Sb$_5$ observed via second-harmonic generation under an in-plane magnetic field. The eMChA signal becomes significant only at temperatures below $T' \approx 35$ K, deep within the charge-ordered state of CsV$_3$Sb$_5$ ($T_{CDW} \approx 94$ K). This temperature dependence reveals a direct correspondence between electronic chirality, unidirectional charge order[7] and spontaneous time-reversal symmetry breaking due to putative orbital loop currents[8–10]. We show that the chirality is set by the out-of-plane field component and that a transition from left- to right-handed transport can be induced by changing the field sign. CsV$_3$Sb$_5$ is the first material in which strong chiral transport can be controlled and switched by small magnetic field changes, in stark contrast to structurally chiral materials, which is a prerequisite for applications in chiral electronics.

The role that symmetries play in determining the properties of matter can hardly be overstated. Two opposite extremes are particularly interesting in crystalline solids. Higher symmetries constrain emergent degrees of freedom to mimic free particles—creating, for instance, massless Dirac or Weyl fermions that recover, at low energies, almost the full Lorentz group of free space[11–14]. A second approach is to study low-symmetry systems with novel responses. Among these, asymmetric systems characterized as 'chiral' play a special role across biology, chemistry and physics[15,16]. Crystals are structurally chiral if they possess no mirror, inversion or roto-inversion symmetry, giving rise to left- and right-handed enantiomers. This chirality can be imprinted on the crystals' emergent excitations, which are then also characterized by a definite handedness. The interaction between structural chirality and the breaking of time-reversal symmetry (TRS) is of particular interest, as it links the static chirality to temporal processes, such as growth, catalysis and wave propagation[17]. Response functions that jointly arise because of chirality and TRS breaking are called magnetochiral anisotropies[18]. Specifically, in metals, one studies the electronic magnetochiral anisotropy (eMChA), which opens up possibilities to detect, manipulate and utilize chiral properties in electronics[1–6].

eMChA usually refers to a change in resistance $R$ due to an applied current $I$ and external magnetic field $B$ that is conventionally expressed as $R(\mathbf{B},\mathbf{I}) = R_0(1 + \mu^2 \mathbf{B}^2 + \gamma^{\pm} \mathbf{B} \cdot \mathbf{I})$[1] (see Fig. 1). Time-reversal symmetry in non-magnetic metals enforces a magnetoresistance even in-field, which usually takes the semi-classical form $\mu^2 \mathbf{B}^2$, with $\mu$ being the mobility. The scalar product $\mathbf{B} \cdot \mathbf{I}$ is only allowed in chiral crystals without space-reflection symmetries, and hence eMChA appears. Its strength is described by the coupling constant, $\gamma^{\pm}$, which takes opposite sign for the two enantiomers and is tensorial in anisotropic conductors. eMChA is most commonly detected by the associated second-harmonic voltage generation under low-frequency a.c. currents, $4V_{2\omega}/V_\omega = \Delta R/R$, where $\Delta R = R(\mathbf{B},\mathbf{I}) - R(\mathbf{B},-\mathbf{I})$ denotes the odd in-current and $R = R(\mathbf{B},\mathbf{I}) + R(\mathbf{B},-\mathbf{I})$ denotes the even in-current contribution to the resistivity.

To display eMChA, a conductor must break inversion symmetry, which can occur as a weak effect in any metal when its macroscopic shape is chiral[2,3], for example, in a coil (Fig. 1). Alternatively, materials with chiral crystal structure[1,4] generally show eMChA in any conductor shape. We note that the 'chiral electronic structure' of the symmetry-broken phase mentioned here does not necessarily have to have the symmetries of a chiral space group. In a layered, quasi-two-dimensional compound one refers to a structure as 'chiral'

[1]Laboratory of Quantum Materials (QMAT), Institute of Materials (IMX), École Polytechnique Fédérale de Lausanne (EPFL), Lausanne, Switzerland. [2]Max Planck Institute for the Structure and Dynamics of Matter, Hamburg, Germany. [3]Centro de Física de Materiales (CSIC-UPV/EHU), Donostia-San Sebastian, Spain. [4]Department of Physics, University of the Basque Country (UPV/EHU), Bilbao, Spain. [5]Donostia International Physics Center, Donostia-San Sebastian, Spain. [6]Fisika Aplikatua Saila, Gipuzkoako Ingeniaritza Eskola, University of the Basque Country (UPV/EHU), Donostia-San Sebastian, Spain. [7]Max Planck Institute for Chemical Physics of Solids, Dresden, Germany. [8]College of Physics, Qingdao University, Qingdao, China. [9]Department of Physics, University of Zürich, Zürich, Switzerland. ✉e-mail: chunyu.guo@mpsd.mpg.de; mark.fischer@uzh.ch; titus.neupert@uzh.ch; philip.moll@mpsd.mpg.de

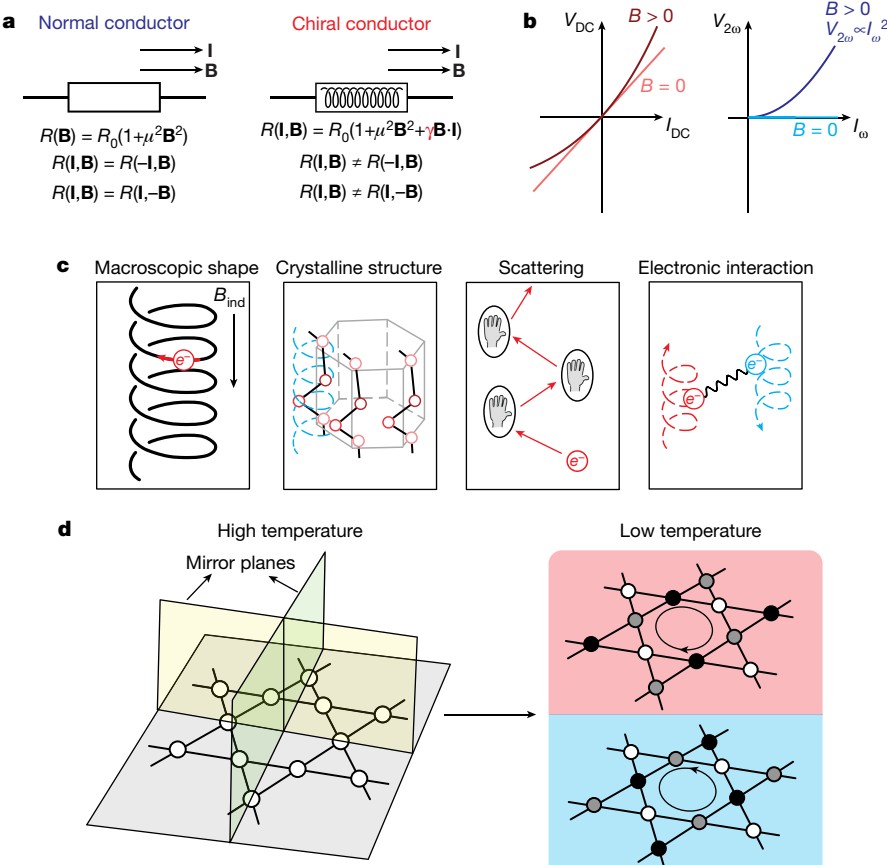

**Fig. 1 | Electronic magnetochiral anisotropy and spontaneous symmetry breaking in CsV₃Sb₅. a**, Illustration of electrical resistance of normal and chiral conductors within the low-frequency limit. **b**, $I(V)$ curve for a chiral conductor. In a d.c. measurement (left), the measured voltage displays a non-linear current dependence with magnetic field applied. In the a.c. case (right), the field-induced second-harmonic voltage, $V_{2\omega}$, depends quadratically on the a.c. current $I_\omega$. **c**, Different mechanisms for electronic magnetochiral anisotropy. The blue dashed line in the crystalline structure case represents the notation of helical atomic chains. For the case of scattering, the encircled hands represent the scattering centres with particular chirality. **d**, The crystal structure of CsV₃Sb₅ preserves all mirror symmetries at high temperatures and only spontaneous symmetry breaking at low temperatures enables a finite eMChA in a symmetric microstructure.

when the in-plane mirror symmetries are broken, whereas the $M_z$ mirror symmetry might still be intact. However, this lack of in-plane mirror symmetry is enough to enable the observation of eMChA in the geometry of our measurements. eMChA expresses an imbalance between scattering processes of different handedness, which can occur either from the intrinsic handedness of the carriers in chiral crystals, or extrinsically from chiral defects, as in plastically twisted conductors. When electronic interactions form ordered phases within chiral materials, as, for example, in chiral magnets, eMChA can be further amplified via scattering off, for example, an emergent chiral spin texture[5,6].

In this work, we demonstrate eMChA in a rectangular bar of CsV₃Sb₅, a layered metal in which vanadium atoms form kagome nets. In this system, a cascade of correlated symmetry-breaking electronic phases emerges at low temperatures[7,10,19–22], including a charge-density wave (CDW) state below $T_{CDW} \approx 94$ K and superconductivity below $T_c \approx 2.5$ K (refs. [19,23–27]). Experimental evidence mounts for a further transition within the charge-ordered phase at $T' \approx 35$ K, accompanied by an additional $4a_0$ unidirectional ordering vector[7] and time-reversal symmetry breaking[8–10]. The sudden onset of unexpectedly strong chiral transport at $T'$ is our main observation. Crucially, this system is centrosymmetric at high temperatures, yet the relevant mirror symmetries are spontaneously broken by correlated phases of the itinerant carriers (Fig. 1). Reversible chirality of the electronic structure within the CDW phase has been observed in scanning tunnelling microscopy (STM) experiments[28]. Note that the accompanying crystal distortion is so weak that the low-temperature crystal structure remains actively debated[7,24,29,30]. In contrast to structurally chiral crystals that strongly differ from their mirror image, here the differences between the enantiomers are subtle and test the limits of experimental resolution. Hence, the observation of eMChA itself in this compound points to its novel origin. As a consequence, the material's chirality itself can be switched, which leads to field-switchable chiral transport in CsV₃Sb₅.

To truly obtain symmetry lowering from spontaneous symmetry breaking, it is critical to avoid any accidental strain fields that may break the symmetry explicitly. To do so, we decouple the crystalline bar mechanically as much as possible from its supporting substrate[31] (Fig. 2a). This structure is mechanically supported by gold-coated SiN$_x$ membrane-based (200 nm thick) springs, and the differential thermal contraction strain is estimated to be less than 20 bar. Any signatures of chiral transport vanish in a reference experiment with even modest strain fields caused by stiff substrate coupling (see Methods), evidencing a strong coupling between the charge order and lattice distortions, which is not surprising in CDW systems[32]. This provides a natural explanation for the opposing STM experiments[28,33].

## Observation of eMChA in CsV₃Sb₅

Our main observation is the appearance of a sizeable second-harmonic response, $V_{2\omega}$, to a low-frequency transport current (7 Hz), which clearly evidences the diode-like behaviour due to chiral transport within the charge-ordered state at low temperatures (Fig. 2). First, we discuss

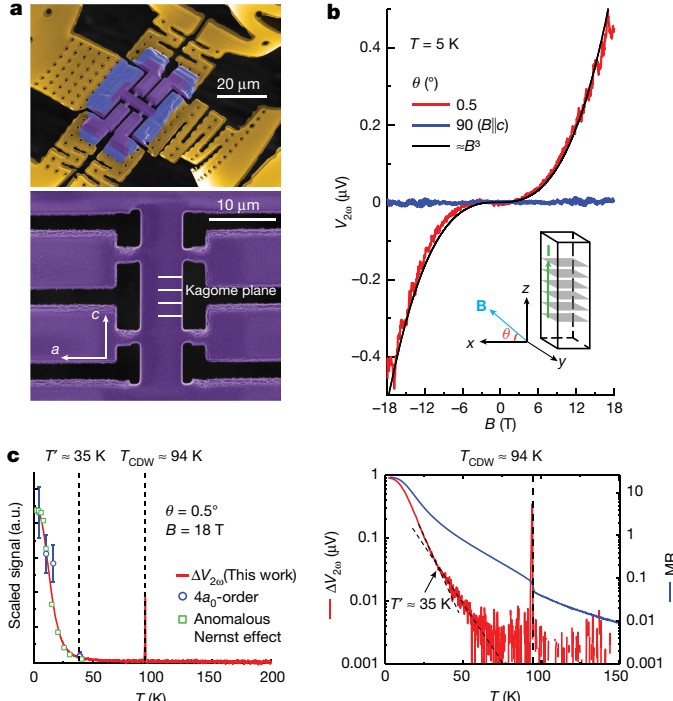

**Fig. 2 | Field and temperature dependence of eMChA. a**, Low-strain microstructure fabricated by focused ion beam. **b**, Field dependence of second-harmonic voltage with current applied along the $z(c)$ axis. The signal becomes sizeable when the magnetic field is applied approximately in the plane. The inset sketches the transport bar. **c**, The red continuous line in the left panel represents temperature dependence of $\Delta V_{2\omega} = V_{2\omega}(18\,\text{T}) - V_{2\omega}(-18\,\text{T})$ with the magnetic field applied approximately in the plane. The blue open circles show the Fourier transform intensity of the wavevector corresponding to the $4a_0$ unidirectional charge-order phase adopted from ref. [7], and the green squares represent the anomalous Nernst effect reported in ref. [34]. The right panel displays log-scale temperature dependence of $\Delta V_{2\omega}$ and magnetoresistance ratio $\text{MR} = (\rho_c(18\,\text{T}) - \rho_c(0))/\rho_c(0)$, where $\rho_c(0)$ denotes the resistivity at zero field.

out-of-plane currents under an approximately in-plane magnetic field, which is purposely misaligned by 0.5° with respect to the kagome planes. At zero field and $T = 5$ K, just above the superconducting transition, no second harmonic is observed, yet the signal quickly grows with increasing magnetic field. Its field dependence is well described by $V_{2\omega} \propto B^3$ up to 18 T, the highest fields accessible to the experiment. This is a striking departure from the behaviour of structurally chiral materials, such as $\alpha$-Te (ref. [1]), where $V_{2\omega}$ displays a linear field dependence, $\Delta R/R = \gamma^{\pm} \mathbf{B} \cdot \mathbf{I}$ (ref. [1]). This suggests that the magnitude of eMChA itself is field dependent, given by $\gamma^{\pm}(\mathbf{B})$.

eMChA depends on the relative direction of the field and the current, and hence, even in a non-linear scenario, $V_{2\omega}(\mathbf{B})$ must change sign when the field polarity reverses, as is observed experimentally. This antisymmetric field dependence provides strong evidence against a putative thermal origin of second-harmonic voltage generation by Joule heating, as the linear magnetoresistance is even in the magnetic field (see Methods). Pronounced quantum oscillations are also observed above $B = 10$ T, demonstrating an influence of Landau quantization on eMChA. This behaviour is observed consistently in two devices with different mechanical mounting approaches, rendering potential torque artefacts due to the soft-mounted structure unlikely. An identically shaped sample probing in-plane transport does not show second-harmonic generation at any field configuration, demonstrating that eMChA is only relevant in the interplane transport (see Methods).

To further characterize eMChA and elucidate its origin in this nearly centrosymmetric material, we next turn to the temperature

dependence of $V_{2\omega}$. Figure 2b displays the raw $V_{2\omega}(\mathbf{B})$ without antisymmetrization. Yet, at elevated temperatures the weak thermal second-harmonic generation can obscure the chiral transport signatures; therefore, we focus on the antisymmetric component $\Delta V_{2\omega} = V_{2\omega}(18\,\text{T}) - V_{2\omega}(-18\,\text{T})$ (see Methods for full data). At high temperatures above $T_{\text{CDW}}$, no $\Delta V_{2\omega}$ is observed, as expected. The transition into the CDW state is clearly evident as a sharp spike in $\Delta V_{2\omega}$ at $T_{\text{CDW}}$, which we associated with the non-analyticity of $R(T)$. A continuous antisymmetric second-harmonic signal only emerges at temperatures below 70 K. Its slow increase with decreasing temperature suddenly accelerates at $T' \approx 35$ K, which is apparent as a change in slope on the logarithmic scale. At lower temperatures, $\Delta V_{2\omega}$ increases significantly and saturates at its maximum value below 3 K. Although our observations only evidence the absence of chiral scattering and do not exclude a chiral order at $T_{\text{CDW}}$ that merely does not affect transport, our results are suggestive of a secondary transition or crossover at lower temperatures of $T'$. In particular, this temperature dependence agrees well with both the Fourier transformation intensity of the $4a_0$ CDW vector ($\mathbf{q}_{4a0}$) obtained from STM experiments[7] and the large anomalous Nernst effect[34]. Such correspondence demonstrates the direct connection between the unidirectional charge order, electronic chirality and hidden magnetic flux. This consistency is further supported by the results of muon-relaxation experiments, which suggest the onset of TRS breaking at around 70 K and a subsequent rearrangement of local field distribution at 30 K (refs. [8–10]).

## Field-switchable electronic chirality

The unusual nature of the eMChA in $CsV_3Sb_5$ becomes apparent when the field orientation is varied with respect to the kagome planes ($\theta = 0°$ denotes the in-plane field orientation; Fig. 3). No $V_{2\omega}$ is observed at large field angles ($\theta > 10°$). Only within a narrow angle range, $\theta = \pm 1°$, does $V_{2\omega}$ quickly grow as the field-angle approaches $\theta = 0°$. It reaches a maximum around $\theta \approx 0.5°$, the configuration discussed previously in Fig. 2. For smaller $\theta$, $V_{2\omega}$ rapidly decreases and vanishes for fields within the kagome planes ($\theta = 0°$). On further rotation, to small negative $\theta$, the signal repeats but with the opposite sign. This marks a most striking aspect of the data: tilting the field across the kagome nets changes the handedness of the material. Rotating the field by 1° barely changes $\mathbf{B}$, hence an abrupt sign change of $V_{2\omega}$ implies a transition into the opposite enantiomer. Furthermore, the signal's magnitude strongly reduces on raising the temperature or lowering the field strength, whereas the angular extent and the sharp anomaly at the in-plane field persists. At temperatures above 35 K the peak is hardly observable, and the faint residual anomaly reflects the exponential drop of $V_{2\omega}$ above $T'$ (Fig. 2d). The rotation curves are slightly hysteric; however, given the sharpness of the steep transition, it was impossible to distinguish an intrinsic hysteresis from the mechanical backlash of the rotator.

The possibility and ease of magnetic manipulation of the electronic chirality presents a unique electromagnetic response of $CsV_3Sb_5$. It suggests that the low-temperature state differs from a simple chiral charge redistribution, as observed, for example, in the $3q$ chiral CDW[35] state of $TiSe_2$. Such a static charge redistribution only couples to magnetic fields via higher order interactions, and its involved lattice response renders it unlikely to be easily manipulated at temperatures well below $T_{\text{CDW}}$. Instead, the experimental situation in $CsV_3Sb_5$ points to coupled TRS breaking, including the concomitant magnetic anomalies at $T_{\text{CDW}}$, the field tunability, as well as muon spectroscopy experiments[8,9]. As a microscopic picture for this correlated state, an orbital loop-current phase in the kagome planes has been proposed, which is consistent with these experimental observations[22,36,37].

## Analysis of eMChA strength

Despite its exotic properties, the eMChA in $CsV_3Sb_5$ can be rationalized within the existing theoretical framework. The magnetoresistance of $CsV_3Sb_5$ is approximately linear in $B$ for small angles $\theta$ at high magnetic

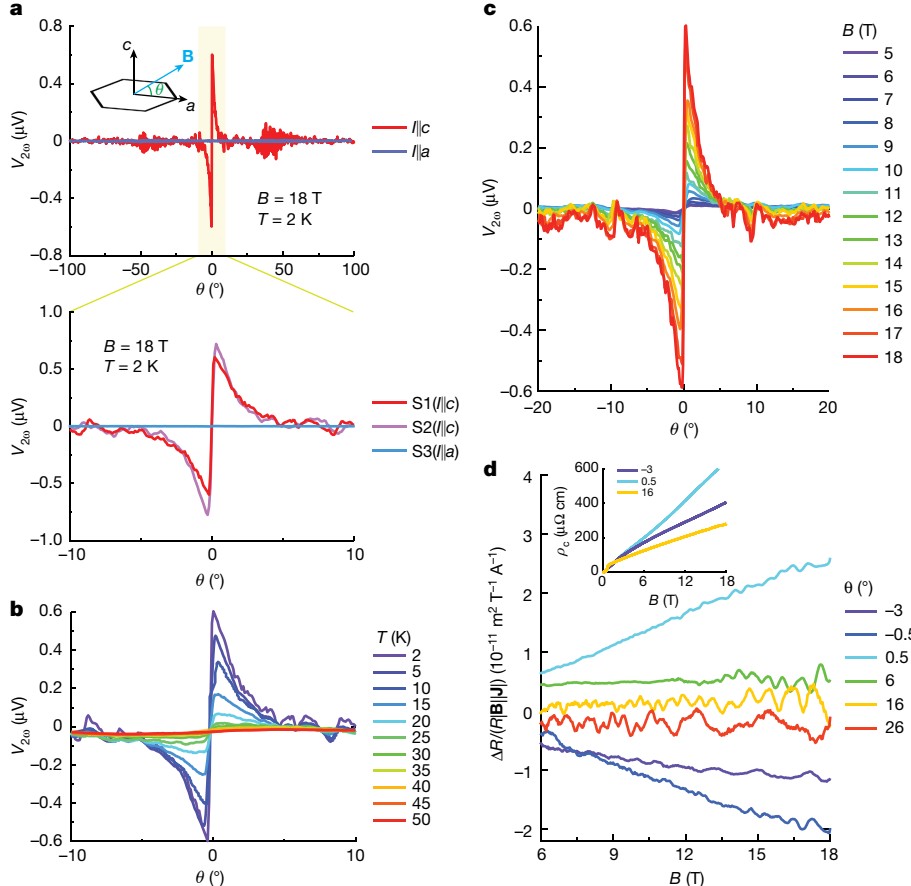

**Fig. 3 | Angular dependence of eMChA. a**, Angular dependence of $V_{2\omega}$ for $I\|a$ and $I\|c$. A sharp spike with sudden sign reversal within 0.5° is observed for $I\|c$ around the $a$ axis. **b,c**, Angular dependence of $V_{2\omega}$ at various temperatures with

fields (see inset of Fig. 3d). Such behaviour is indeed not unexpected for a material with density-wave order[38] and has also been observed in many semimetals[39,40]. This marks a crucial difference to previous eMChA studies, in which the conventional resistance $R(\mathbf{B},\mathbf{I}) + R(\mathbf{B},-\mathbf{I}) \approx 2R_0$ remains approximately field independent. This, and the strong $\theta$ dependence of the eMChA coefficient $\gamma$, means that eMChA cannot be characterized by a constant tensor, as is common practice in the literature for conventional eMChA materials. Yet, we can gain some insights about the magnitude of eMChA by computing $\Delta R/(R|\mathbf{B}\|\mathbf{J}|) = 4V_{2\omega}/(V_\omega|\mathbf{B}\|\mathbf{J}|)$ for given magnetic field strength $|\mathbf{B}|$ and current density $\mathbf{J}$ (see Fig. 3d)[1], for quantitative comparisons to other systems. The quantity $\Delta R/(R|\mathbf{B}\|\mathbf{J}|)$ equals the constant $\gamma$ when it is a field-independent parameter in chiral materials commonly used in the literature. At $B = 18$ T and $\theta = 0.5°$ we find $\Delta R/(R|\mathbf{B}\|\mathbf{J}|) \approx 2.4 \times 10^{-11}$ m² T⁻¹ A⁻¹. In comparison, this value is smaller than its record observations in t-Te (ref. [1]) (10⁻⁸ m² T⁻¹ A⁻¹) and TTF-ClO₄ (ref. [4]) (10⁻¹⁰ m²; T⁻¹ A⁻¹), for which the distinct structural chirality results in relatively large eMChA, whereas it is larger than that of chiral magnets, such as CrNb₃S₆ (ref. [5]) (10⁻¹² m² T⁻¹ A⁻¹) and MnSi (ref. [6]) (10⁻¹³ m² T⁻¹ A⁻¹), in which the chiral spin texture plays a major role in eMChA.

As the conventional eMChA analysis is only applicable for materials with negligible magnetoresistance, a description in terms of the conductance is more appropriate to further capture the lowest order field-tuned behaviour of the response in CsV₃Sb₅ (see Methods). For purely longitudinal transport and negligible Hall resistivity, the conductance is the inverse of the resistance, such that in analogy to the usual analysis of eMChA, for the conductance we write

$B = 18$ T (**b**) and of various magnetic fields at $T = 2$ K (**c**). **d**, Field-dependent eMChA coefficients at various angles. The inset displays the magnetoresistance measured with current applied along the $c$ axis at various field directions.

$\sigma + \Delta\sigma \approx 1/R - \Delta R/2R^2$ (see Methods). We can thus extract $\Delta\sigma \propto V_{2\omega} / V_\omega^2$, where $V_\omega$ is now linear in $B$ for large fields. For a field applied approximately in-plane, $\Delta\sigma$ is thus approximately linear in $B$, which is the lowest order coupling between magnetic field and current (see Methods) and naturally explains the $B^3$ dependence of $V_{2\omega}$. The linear field dependence of $\Delta\sigma$ yields a field-independent first-order derivative $\partial(\Delta\sigma)/\partial B$ (see Fig. 4). The sudden sign reversal of $\partial(\Delta\sigma)/\partial B$ for small $\theta$ then suggests that the out-of-plane component of the field, $B_z$, has a non-perturbative effect on the system and we treat it separately, whereas the in-plane component is a perturbation to linear order. In other words, we write $\Delta\sigma(\mathbf{B}, I_z) = \tilde{\sigma}(B_z)B_x I_z$. Note that such a coupling is only allowed for a system that breaks the $y \mapsto -y$ mirror symmetry. With $\Delta\sigma(\mathbf{B},I_x)$ vanishingly small, no similar conclusion can be drawn for the mirror symmetry $z \mapsto -z$.

## Theoretical modelling

The behaviour of $\Delta\sigma$ seen in our experiment demonstrates that the charge order in CsV₃Sb₅: (1) breaks in-plane mirror symmetries, at least below $T' \approx 35$ K and (2) can be manipulated by an out-of-plane magnetic field in the same temperature regime. We thus establish that the tunability of the chirality of charge order in CsV₃Sb₅, previously seen in STM experiments, is a macroscopic bulk property of the unconventional charge order.

We further propose the following qualitative scenario, which would be consistent with the full $\theta$ dependence of our experimental observations and calls for confirmation by local-probe techniques (Fig. 4). $B_z$ is the natural tuning parameter in kagome-net physics, as evidenced by

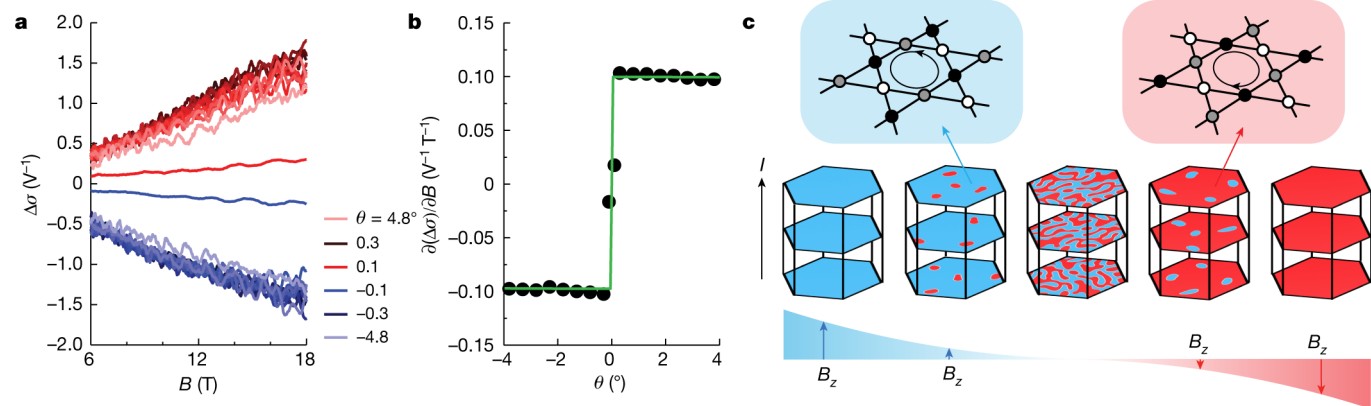

**Fig. 4 | Theoretical modelling of field-switchable chiral domains.**
**a**, Field dependence of chiral conductivity $\Delta\sigma$ at various angles. Between $\theta = \pm 0.3$ and $\pm 4.8°$ all data are measured with an angle step of $0.5°$. **b**, Angular dependence of the first-order derivative $\partial(\Delta\sigma)/\partial B$ from $B = 6$ to 18 T. The green curve represents the model description of chiral conductivity:

$\frac{\partial\Delta\sigma}{\partial B} = \text{sign}(\theta)\,\cos(\theta)\,\frac{\Delta_0 \bar{M} e^2 \tau}{2\pi^3 \hbar}$, as derived in the Methods. **c**, Sketch of a chirality reversal at in-plane aligned magnetic fields. The emergence of opposite domains can naturally lead to a strong enhancement of eMChA at low field angles.

our experiments as well as STM experiments. Akin to a soft ferromagnet, large values of $B_z$ (large $\theta$) induce a fully polarized, monochiral state. In this polarized state, only intrinsic chiral scattering processes induce eMChA, which commonly are weak. As the field is tilted towards the plane, $B_z$ is reduced and domains of opposite chirality appear, which act as ideal chiral scattering centres. Hence, domain-wall scattering leads to strong extrinsic eMChA. Naturally, a local probe, such as STM, would observe a chiral structure and occasionally the required domain boundaries between them, as indeed is the experimental situation[28,33]. At even smaller $B_z$ for fields very close to the planes, both chiralities appear symmetrically and hence a globally averaging probe, such as transport, observes a macroscopically symmetric conductor with vanishing eMChA. A fully symmetric process appears if the field is turned further, yet with inverted roles of majority and minority chirality.

In this scenario, the chirality switching is driven by $B_z$ independent of the in-plane field, in particular it would also occur for fully out-of-plane fields, where no eMChA is observed in our experiment. Yet, unlike structurally chiral systems, here the magnetic field plays a dual role. Whereas $B_z$ sensitively changes the sign of $\partial(\Delta\sigma)/\partial B$, the large in-plane field is essential to observe finite eMChA, as $\Delta\sigma \propto I_z|\mathbf{B}|$. Given the close relationship between the field-switchable chiral transport and the chiral domains in $CsV_3Sb_5$, it is worth exploring its generality in other materials with suspected chiral orbital loop current.

## Outlook

Although the small magnitude and extreme environmental conditions probably preclude direct applications of $CsV_3Sb_5$, it showcases that spontaneous symmetry breaking can be used to transform small changes in external fields into singular changes in the response functions of chiral conductors. Given the subtle deviation from centrosymmetry of the charge-ordered phase, the emergence of eMChA in correlated states calls for new theoretical approaches to identify the microscopic mechanisms. The multitude of competing ground states in correlated materials gives rise to their versatility and tunability, which now presents a new approach towards chiral transport. In this direction, the field-switchable chiral transport adds a new aspect to the emergent picture of a highly frustrated, strongly interacting electron system on the kagome planes of $CsV_3Sb_5$. Although the magnitude of eMChA is unexpectedly large, these results link well with recent works that have shown the charge-ordered state to be chiral and TRS breaking[8–10,28]. Akin to the theme of coupled orders in multiferroics, a series of new response functions emerges in materials such as $CsV_3Sb_5$, with multiple intertwined order parameters.

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

# Methods

## Crystal synthesis and characterization

$CsV_3Sb_5$ crystallizes in the $P6/mmm$ space group, which features a layered structure of kagome planes formed by the V atoms (Extended Data Fig. 1). The single crystals were grown by the self-flux method[24]. Hexagonal plate-shaped crystals with typical dimensions of $2 \times 2 \times 0.04$ mm$^3$ were obtained. The crystals were characterized by X-ray diffraction off the maximum surface on a PANalytical diffractometer with CuK$\alpha$ radiation at room temperature. As shown in Extended Data Fig. 1, all the peaks in the X-ray diffraction pattern can be identified as the (00$l$) reflections of $CsV_3Sb_5$.

On the basis of the crystalline structure, we have calculated the band structure of $CsV_3Sb_5$ by density functional theory using the Quantum Espresso package[41], the details of which can be found in ref. [31]. The obtained electronic structure features multiple Dirac nodal lines lifted by spin-orbit coupling, leaving only symmetry-protected Dirac nodes at L points. These results are consistent with previous reports[23,24].

Magnetoresistivity measurements were performed with electric current and magnetic field applied along out-of-plane ($z$) and in-plane ($x$) directions, respectively. The magnetoresistance displays a quasi-linear field dependence up to $B = 18$ T, whereas Hall resistivity is almost negligible compared with that. This is expected, as the electrical current is applied along the out-of-plane direction the Hall resistivity should vanish for such a quasi-two-dimensional material, with the Brillouin zone dominated by the large cylindrical Fermi surfaces.

## Angular dependence of magnetoresistance and its relation to $V_{2\omega}$

At low temperature, angle-dependent magnetoresistivity displays a strong peak when the magnetic field direction rotates across the kagome plane (Extended Data Fig. 2). With increasing temperature this peak gradually transitions to a broad hump at $T = 50$ K. This strong enhancement of magnetoresistivity at in-plane fields is probably a feature of open-orbit magnetotransport, which is expected for a metal featuring cylindrical Fermi-surface sheets[42]. The smearing of the spike at high temperatures therefore demonstrates the reduction of carrier mobility with increasing temperature.

In the meantime, the strong increase of magnetoresistivity naturally leads to enhancement of $V_{2\omega}$. Combining the angular dependence of both leads to a clear quadratic relationship between $V_{2\omega}$ and $\rho_c$, which provides further evidence for the chiral conductance analysis.

## $I$–$V$ characteristics of eMChA

Second-harmonic voltage generation due to eMChA is expected to display a quadratic current dependence. Here we present the $I$–$V$ characteristics of both first- and second-harmonic voltages measured with a 7 Hz a.c. current (Extended Data Fig. 3). For $V_\omega$ the relationship depends linearly on current, which corresponds to the first-order resistance term. On the other hand, the second-harmonic voltage shows a clear quadratic current dependence, which is an expected signature of eMChA. These results again demonstrate that the observed $V_{2\omega}$ originates from the chiral correction of conductivity due to the electronic chirality of $CsV_3Sb_5$.

## Examination of Joule heating effect

Joule heating is a natural extrinsic origin of higher harmonic voltage generation[43,44]. Applying an a.c. electric current, $I_\omega$, must result in an oscillating temperature with a frequency of $2\omega$. Therefore if the electrodes of the device are strongly imbalanced in contact resistance, an extrinsic $V_{2\omega}$ can be observed. To further check the influence of Joule heating in the measurements of eMChA in $CsV_3Sb_5$, we performed systematic current-dependent $V_{2\omega}$ measurements under different thermal conditions (Extended Data Fig. 4). By controlling the helium gas pressure of the sample space, the thermal link between the device and the sample chamber can be easily tuned. Within a low current regime (below 0.12 mA), the collapse of all curves measured at different conditions suggest the insignificance of the Joule heating effect. With further increasing current, Joule heating inevitably grows and becomes detectable. To avoid any disturbance due to Joule heating, all measurements of eMChA have been performed with a relatively low a.c. current of 0.1 mA and high gas pressure ($p_s \approx 600$ mbar) of the sample space, providing the maximal cooling power.

## Reproducibility of eMChA with two different devices

To show the reproducibility of the second-harmonic voltage generation due to eMChA in $CsV_3Sb_5$, we have measured two membrane-based devices with different mounting techniques/geometries (Extended Data Fig. 5). For device S1, the sample was completely suspended by soft Au-coated membrane springs. In comparison, device S2 was attached to the membrane only on one side, and the other side of the sample was welded directly to the Si substrate by focus ion beam (FIB)-assisted Pt deposition. Device S2 displays a slightly broader CDW transition than S1 in the temperature dependence of resistivity across $T_{CDW}$, yet the transition temperatures are exactly the same. This suggests a marginally larger strain gradient across the device due to thermal contraction for device S2, which is compatible with the estimated strain value presented in the next section. The second-harmonic voltage was consistently observed among the two devices, with a similar value as well as an almost identical angular spectrum. These results demonstrate the clear consistency among different low-strain samples, and therefore evidence that the observed eMChA in $CsV_3Sb_5$ is an intrinsic material property. These devices further differ in their coupling strength between the substrate and the device. In view of the much stiffer coupling in S2, the similarity of the data speaks against magnetic torque induced angle changes as a putative error source.

## Estimation of strain due to differential thermal contraction

To obtain the tensile strain applied to the sample, we first need to estimate the total displacement of the samples and substrates used due to different thermal contraction coefficients. On cooling from 300 K to 4 K, the integrated thermal contraction coefficients of SiN$_x$ ($\varepsilon_{SiN}$) and Si ($\varepsilon_{Si}$) were 0.0342% and 0.0208%, respectively. For the sample itself we assumed a typical thermal contraction coefficient for alkali metal of $\varepsilon_{Sample} \approx 0.1\%$, which provides a conservative, upper bound. On the basis of these parameters and the actual device geometry illustrated in Extended Data Fig. 6, the total displacement can be easily obtained as:

$$dL_{S1} = L_{S1} \times \varepsilon_{Sample} = 30 \text{ nm} \tag{1}$$

$$dL_{SiN_x} = L_{SiN_x} \times \varepsilon_{SiN_x} = 75 \text{ nm} \tag{2}$$

$$dL_{Si} = L_{Si} \times \varepsilon_{Si} = 52 \text{ nm} \tag{3}$$

$$dL_{S1} = dL_{S1} + dL_{SiN_x} - dL_{Si} = 53 \text{ nm} \tag{4}$$

The spring constant of the SiN$_x$ spring for device S1 is estimated as 125 N m$^{-1}$ from finite element simulations[31] (COMSOL), the total pressure can be calculated as:

$$P_{S1} = k_{S1} \cdot dL_{S1}/A = 8.8 \text{ bar} \tag{5}$$

where $A$ is the cross-section of the spring.

Meanwhile, for device S2, the pressure can be calculated using the same process:

$$P_{S2} = k_{S2} \cdot dL_{S2}/A = 18.7 \text{ bar} \tag{6}$$

In both cases the pressure is less than 20 bar. Taking the typical Young's modulus of alkali metals ($\approx 5$ GPa), the strain applied on the sample is estimated to be $\approx 0.04\%$, which quantifies the low-strain nature of these devices.

## Strain effect on eMChA

The necessity of low-strain mounting was revealed by a comparative study of device S4, which features a sample that is glued down to a sapphire substrate. Here the device is structured into an L shape with two long beams along both the $a$ and $c$ directions (Extended Data Fig. 7). As the sample and substrate are mechanically coupled via the glue droplet, the thermal contraction difference between them results in a tensile strain along the beam direction. This tensile strain not only shifts the CDW transition of device S4 to a higher temperature compared to the strain-free S1, but also suppresses the superconducting transition down to lower temperatures. Most importantly, no meaningful second-harmonic voltage has been observed for device S4. These observations suggest the importance of $c$ axis tensile strain, which in defining an extrinsic, long-range domain structure, is unable to be switched or tuned. This observation suggests that residual strain fields would provide a natural explanation for the contradictory STM experimental results[28,33].

## Field-symmetry analysis of second-harmonic voltage

To further demonstrate the origin of second-harmonic voltage generation, we also measured the temperature-dependent $V_{2\omega}$ at $B = 18$ and $-18$ T (Extended Data Fig. 8). By taking the sum and difference of these two results we obtained both the field-symmetric and -asymmetric components of $V_{2\omega}$. It is clear that the antisymmetric component dominates the total signal at low temperatures, whereas the symmetric component, which is probably due to Joule heating at the electric contacts, is merely a minor part.

## Theoretical considerations

In this section, we discuss the magnetoresistance of a single-band model to illustrate the appearance of the various contributions to the linear magnetoresistance, and to the second-order response discussed in the main text. In particular, we are interested in the effect of a CDW on the magnetotransport, when the CDW not only breaks translation, but also breaks time-reversal and several mirror symmetries. Note that we focus here on intrinsic contributions that enable us to explain the abrupt switching of the second-order response at small $\theta$. To model the full $\theta$ dependence, extrinsic contributions would have to be included as well, as discussed in the main text.

In most metals, the transverse magnetoresistance scales quadratically with magnetic field, $\rho_{zz}(B_x) \propto B_x^2$. However, this behaviour can change to linear with $B$ if there are small Fermi surfaces or Fermi surfaces with sharp corners[45]. Although the conditions for such $B$-linear behaviour are probably not satisfied in the normal state of $CsV_3Sb_5$, the Fermi-surface reconstruction due to the CDW instability is expected to result in new, smaller Fermi surfaces, such that a linear magnetoresistance, as observed, can be explained. Note that linear magnetoresistance in density-wave materials has indeed been observed and discussed in the context of Fermi-surface reconstruction by Feng and co-workers[38].

When the density-wave instability breaks additional symmetries, we find further contributions to the magnetoresistance, or, equivalently, to the conductivity. To see this, we use Boltzmann transport theory in the relaxation-time approximation, in which the conductivity is given by

$$\sigma_{ij} = \frac{e^2 \tau}{4\pi^3 \hbar^2} \int d^3 k v_i(\mathbf{k}) v_j(\mathbf{k}) \frac{\partial f(\xi)}{\partial \xi} \tag{7}$$

with $e$ the electron charge, $\tau$ the scattering time and

$$\hbar \boldsymbol{v}(\mathbf{k}) = \nabla_{\mathbf{k}} \xi_{\mathbf{k}} \tag{8}$$

the velocity for electrons with dispersion $\xi_{\mathbf{k}}$. We assume in the following that we are in the symmetry-broken charge-ordered phase and the dispersion is given by $\xi_{\mathbf{k}}^{CDW}$ at zero magnetic field. If we minimally cou-

ple the vector potential to the momentum, we first find the dominant term $\sigma_{zz}^0 \propto 1/|B|$, as discussed above.

In addition to minimal coupling, a magnetic field can couple to the electron dispersion directly, if its Bloch states have an orbital magnetic moment $\mathbf{M}(\mathbf{k})$. In this case, we can write

$$\xi_{\mathbf{k}} = \xi_{\mathbf{k}}^{CDW} + \mathbf{M}(\mathbf{k}) \cdot \mathbf{B}. \tag{9}$$

Note that $\mathbf{M}(\mathbf{k})$ is a pseudovector. If TRS is present, $\mathbf{M}(\mathbf{k})$ is an odd function of $\mathbf{k}$. If in addition inversion symmetry is present, we find $\mathbf{M}(\mathbf{k}) \equiv 0$. This should be the case above the CDW transition temperature. In the CDW phase, by contrast, the various broken symmetries enable $\mathbf{M}(\mathbf{k})$ to be non-vanishing. For concreteness, we will now only discuss the case of broken $x$ mirror symmetry, which is sufficient to explain the experimentally observed response. In this case, we find to lowest order in $\mathbf{k}$ that $M_x(\mathbf{k}) \propto k_z$. If, further, this symmetry breaking is due to an ordered phase with order parameter $\Delta^{CDW}$, we can expand in both $\mathbf{k}$ and $\Delta^{CDW}$ and write $M_x(\mathbf{k}) \approx \widetilde{M} \Delta^{CDW} k_z$ with $\widetilde{M}$ a model-dependent constant.

We can now use eqns (8) and (9) to calculate the velocity in the $z$ direction,

$$\hbar v_z(\mathbf{k}) = v_z^0(\mathbf{k}) + \frac{\partial \mathbf{M}(\mathbf{k})}{\partial k_z} \cdot \mathbf{B} \approx v_z^0(\mathbf{k}) + \Delta^{CDW} \widetilde{M} B_x. \tag{10}$$

(More generally, $M_x(\mathbf{k})$ will be an odd function of $k_z$, such that the additional contribution to the velocity will be even.) Finally, we find for the conductivity in the $z$ direction (dropping, for simplicity, the subscripts $zz$)

$$\sigma(\mathbf{B}, \mathbf{I}) \approx \sigma + \frac{\Delta^{CDW} \widetilde{M} B_x e^2 \tau}{2\pi^3 \hbar} \int d^3 k v_z^0(\mathbf{k}) \frac{\partial f(\xi)}{\partial \xi} \tag{11}$$

$$\approx \sigma + \frac{\Delta^{CDW} \widetilde{M} e^2 \tau}{2\pi^3 \hbar} B_x I_z, \tag{12}$$

up to the order of $B_x(\Delta^{CDW})^2$. Here we have used the stationary Fermi distribution to calculate the current carried by the system. We thus find an additional contribution to the conductivity $\Delta\sigma \propto B_x I_z$, which will result in a second-harmonic signal in an a.c. electric field applied in the $z$ direction.

To further support the above arguments for the case of $CsV_3Sb_5$, we have performed a tight-binding calculation, which shows, in the chiral CDW phase, that: (1) the Fermi-surface structure does indeed become more structured with sharp corners, and (2) a finite orbital magnetic moment $M_x$ arises, see Extended Data Fig. 9.

Note that, experimentally, we find that a small magnetic field in the $z$ direction can change the sign of the observed signal. This implies that a magnetic field $B_z$ couples linearly to the order parameter $\Delta^{CDW}$, which in turn implies that the order breaks TRS in addition to the mirrors $M_x$ and $M_y$. This is in agreement with the experimental findings in ref. [28]. In terms of a simple Landau theory,

$$F[\Delta^{CDW}] = \alpha(\Delta^{CDW})^2 + \frac{\beta}{2}(\Delta^{CDW})^4 + \gamma B_z \Delta^{CDW}, \tag{13}$$

with $\alpha < 0$, $\beta > 0$ and $\gamma \ll |\alpha|$, $\beta$, the CDW order parameter is $\Delta^{CDW} \approx \text{sign}(B_z)\sqrt{-\alpha/\beta} = \text{sign}(\theta)\Delta_0$, with $B_z = |\mathbf{B}|\sin\theta$, implying its sign change with $B_z$.

In summary, for the slope of the chiral conductivity, we find

$$\frac{\partial \Delta\sigma}{\partial B} = \text{sign}(\theta) \cos(\theta) \frac{\Delta_0 \widetilde{M} e^2 \tau}{2\pi^3 \hbar}. \tag{14}$$

For small angles $\theta$ off the basal plane, this yields a step function in good qualitative agreement with the experimentally extracted form for

small angles $\theta$. Moreover, combining the angular dependence of both the theoretically predicted chiral conductivity and the experimentally measured magnetoresistance, we also derived the second-harmonic voltage $V_{2\omega}$ as a function of angle (Extended Data Fig. 10), which is also consistent with the experimental results.

Finally, if we want to compare the conductivity calculated above to the experiment and the standard eMChA literature, we need to express the conductivity in eqn (12) as a resistance. Namely, one usually writes $R(\mathbf{B}, \mathbf{I}) = R + \Delta R/2$, where in general $R$ can depend on $B$, and in particular here we have $R \approx |B|$. We thus find

$$\sigma(\mathbf{B}, \mathbf{I}) = 1/R(\mathbf{B}, \mathbf{I}) \approx 1/R - \Delta R/2R^2. \tag{15}$$

This yields $\Delta\sigma \approx -\Delta R/2R^2$ and in turn, we expect

$$\Delta R \approx 2\Delta\sigma R^2 \tag{16}$$

By applying a low-frequency a.c. current $I_\omega = I_0\sin(\omega t)$, the generated electric voltage can be expressed as:

$$
\begin{aligned}
V &= I_\omega(R + \Delta R/2) \\
&\approx I_\omega R + \tilde{\sigma}BR^2I_\omega^2 \\
&= I_0R\sin(\omega t) + \frac{\tilde{\sigma}BR^2I_0^2}{2} - \frac{\tilde{\sigma}BR^2I_0^2}{2}\cos(2\omega t) \\
&= V_\omega\sin(\omega t) + V_{DC} - V_{2\omega}\cos(2\omega t),
\end{aligned}
\tag{17}
$$

here $V_{DC}$ stands for the d.c. background voltage. This yields:

$$V_{2\omega}/V_\omega = \frac{\tilde{\sigma}BRI_0}{2} = \frac{1}{4}\frac{2\Delta\sigma R^2}{R} = \frac{1}{4}\frac{\Delta R}{R}, \tag{18}$$

which also suggests:

$$V_{2\omega} \propto BR^2 \propto B_x^3, \tag{19}$$

exactly in line with our experimental data.

## Data availability

Data that support the findings of this study are deposited to Zenodo with the access link: https://doi.org/10.5281/zenodo.6787797.

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

**Acknowledgements** This work was funded by the European Research Council (ERC) under the European Union's Horizon 2020 research and innovation programme (MiTopMat, grant agreement no. 715730, and PARATOP, grant agreement no. 757867). This project received funding by the Swiss National Science Foundation (grant no. PP00P2_176789). M.G.V., I.E. and M.G.-A. acknowledge the Spanish Ministerio de Ciencia e Innovacion (grant PID2019-109905GB-C21). M.G.V., C.F. and T.N. acknowledge support from FOR 5249 (QUAST) lead by the Deutsche Forschungsgemeinschaft (DFG, German Research Foundation). This work has been supported in part by Basque Government grant IT979-16. This work was also supported by the European Research Council Advanced Grant (no. 742068) 'TOPMAT', the Deutsche Forschungsgemeinschaft (Project-ID no. 247310070) 'SFB 1143' and the DFG through the Würzburg–Dresden Cluster of Excellence on Complexity and Topology in Quantum Matter ct.qmat (EXC 2147, Project-ID no. 390858490).

**Author contributions** Crystals were synthesized and characterized by D.C. and C.F. The experiment design, FIB microstructuring, the magnetotransport measurements and the second-harmonic voltage measurements were performed by C.G., C.P., S.K., X.H. and P.J.W.M. M.H.F. and T.N. developed and applied the general theoretical framework, and the analysis of experimental results has been done by C.G., C.P. and P.J.W.M. Band structures were calculated by M.G.-A., I.E. and M.G.V. All authors were involved in writing the paper.

**Funding** Open access funding provided by Max Planck Society.

**Competing interests** The authors declare no competing interests.

**Additional information**
**Correspondence and requests for materials** should be addressed to Chunyu Guo, Mark H. Fischer, Titus Neupert or Philip J. W. Moll.

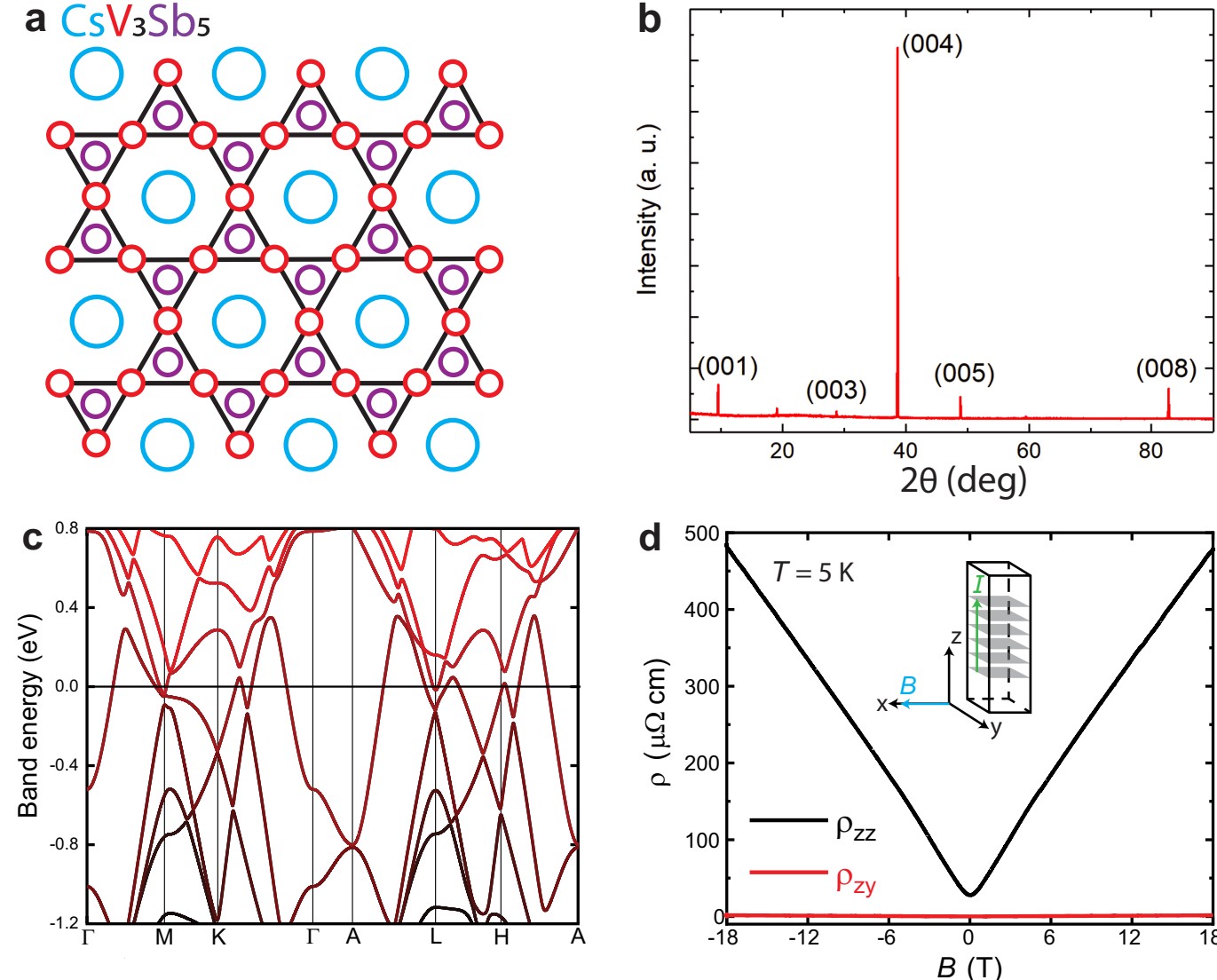

**Extended Data Fig. 1 | Basic properties of CsV₃Sb₅. a**, Crystal structure of CsV₃Sb₅. **b**, XRD pattern of the (001) facet of a CsV₃Sb₅ crystal. **c**, Band structure of CsV₃Sb₅ calculated by density functional theory (DFT) using the Quantum Espresso package (QE)[41]. **d**, Field dependence of magnetoresistivity and Hall resistivity measured at $T = 5$ K. A large quasi-linear magnetoresistance is observed up to $B = 18$ T. In comparison, the Hall resistivity is almost negligible.

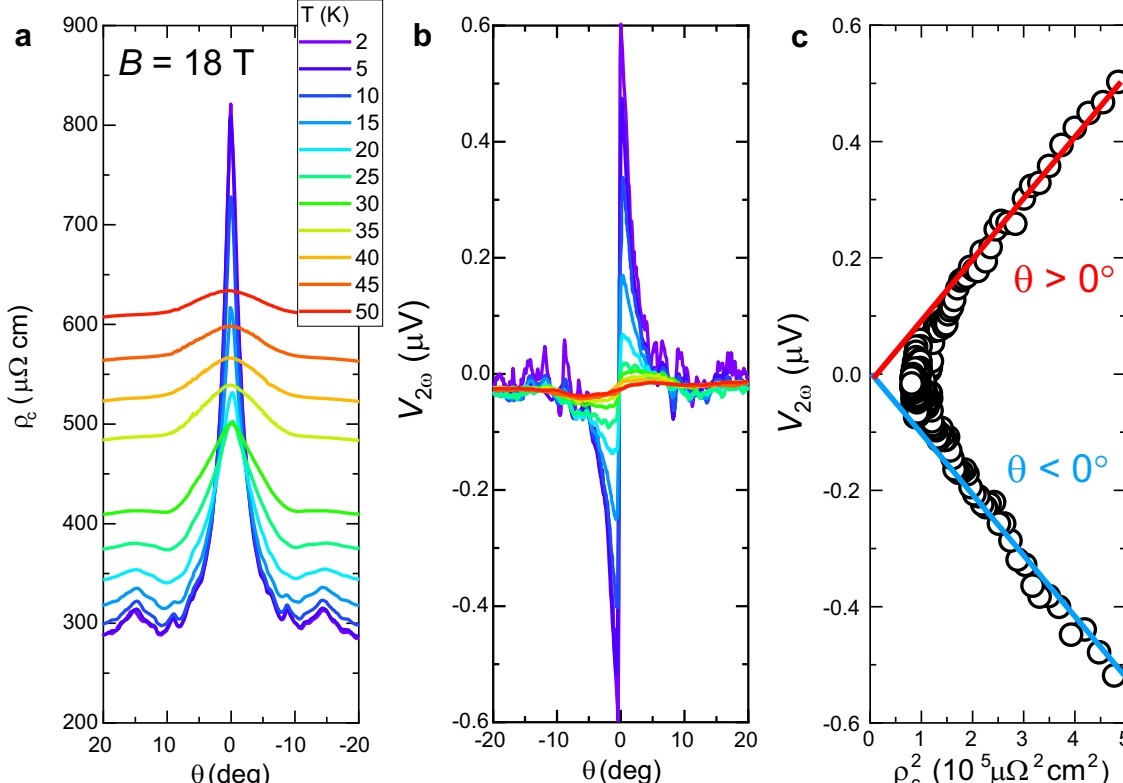

**Extended Data Fig. 2 | Angular dependence of magnetoresistivity and eMChA. a**, Angle-dependent magnetoresistivity of $CsV_3Sb_5$ measured with $B = 18$ T at various temperatures. A strong spike can be observed around $\theta = 0$ deg which becomes broader with increasing temperatures. **b**, Correspondingly, $V_{2\omega}$ also gets pronounced within the same angle range. **c**, The $V_{2\omega}$ depends linearly on the square of c-axis resistivity, which demonstrates the direct connection between magneto-resistivity and eMChA.

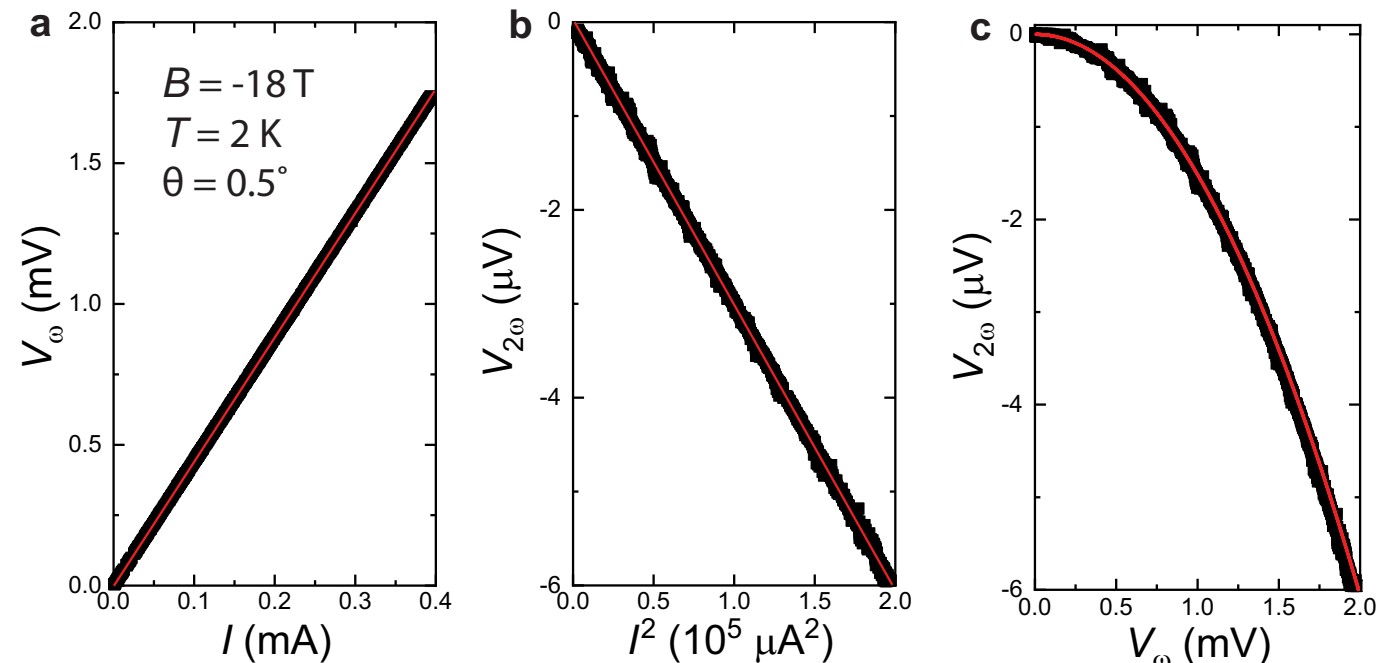

**Extended Data Fig. 3 | Current dependence of first and second harmonic voltage. a, b,** Current-dependence of both $V_\omega$ and $V_{2\omega}$. As expected $V_\omega$ depends linearly on current while $V_{2\omega}$ displays a quadratic current dependence. **c,** Summary of the relation between $V_{2\omega}$ and $V_\omega$, which shows a parabolic dependence.

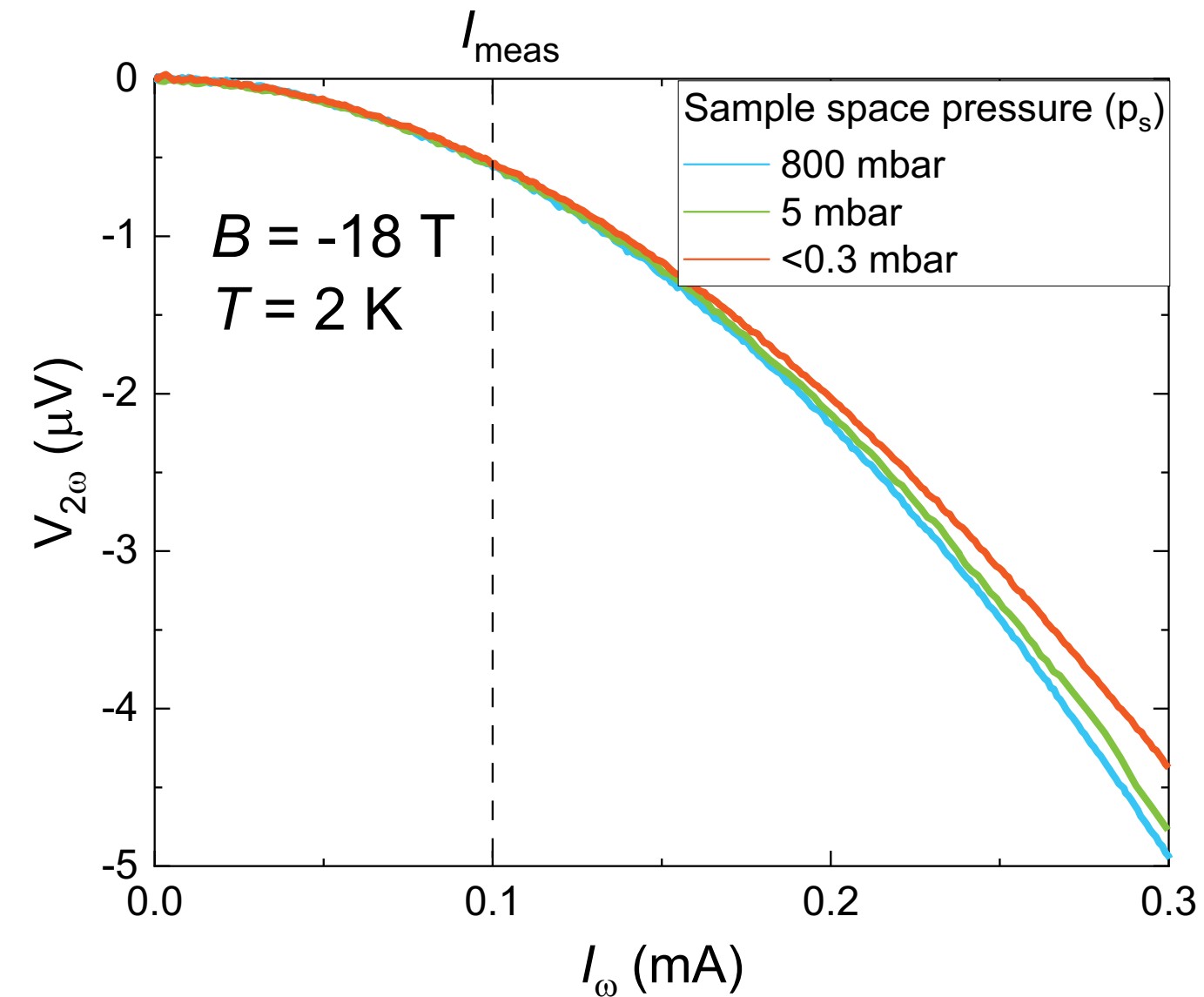

**Extended Data Fig. 4 | Influence of Joule heating effect.** Current-dependence of $V_{2\omega}$ measured at $B = 18$ T and $T = 2$ K with varying levels of helium exchange gas pressure in the cryostat. The curves differ only at currents above 0.12 mA, suggesting that the heat generation and accumulation at lower currents is not a dominant factor. eMChA was measured at lower values of 0.1 mA.

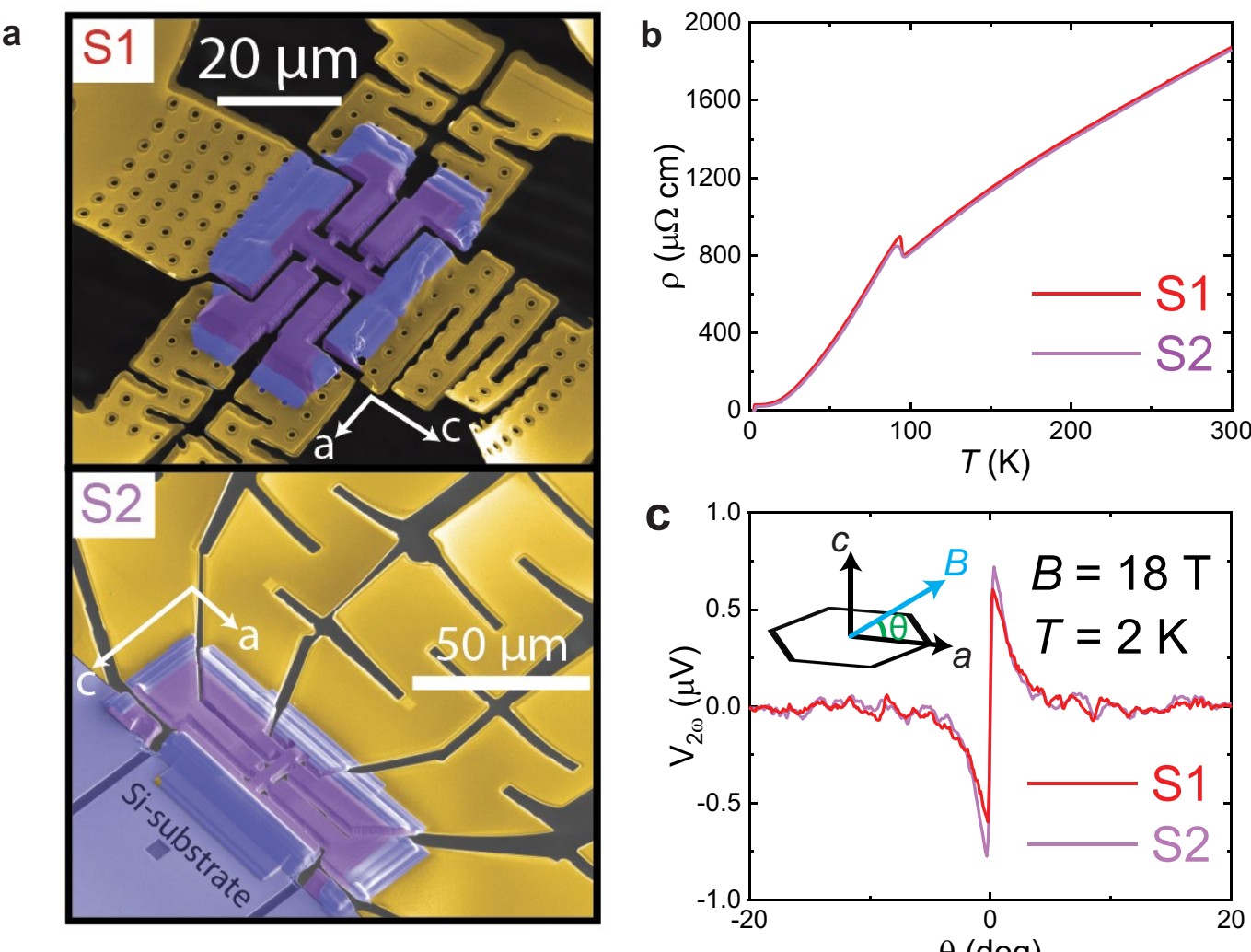

**Extended Data Fig. 5 | Reproducibility of eMChA with two different devices. a**, Scanning electron microscope (SEM) images of devices S1 and S2. **b**, Temperature-dependent resistivity of S1 and S2 from 300 K to 1.6 K. **c**, Angle-dependent second harmonic voltage measured at $B = 18$ T and $T = 2$ K.

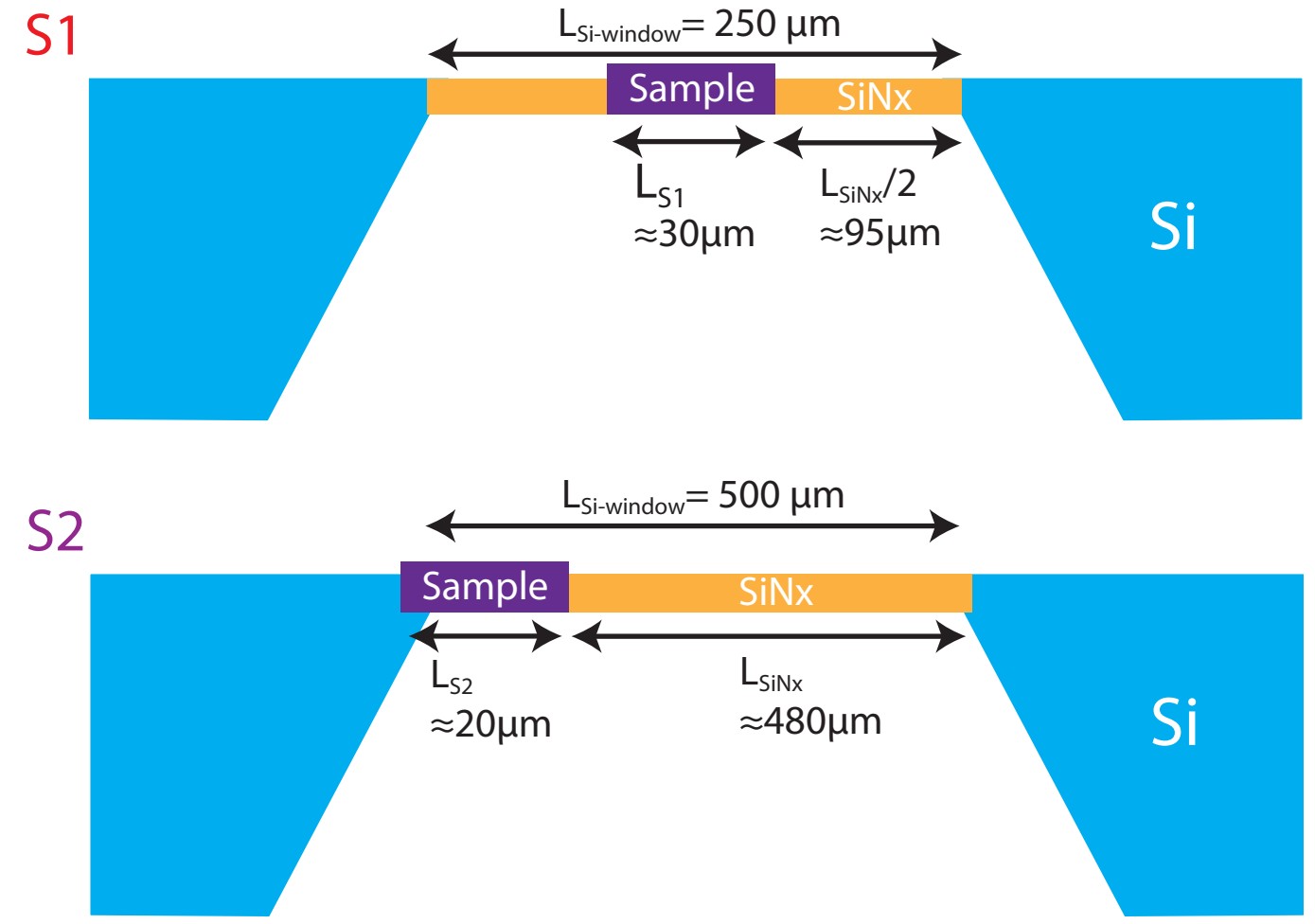

**Extended Data Fig. 6 | Illustration of device configuration for both S1 and S2.** While S1 is completely suspended by the membrane springs, S2 is attached to the Si-substrate frame on one side and membrane springs on the other side.

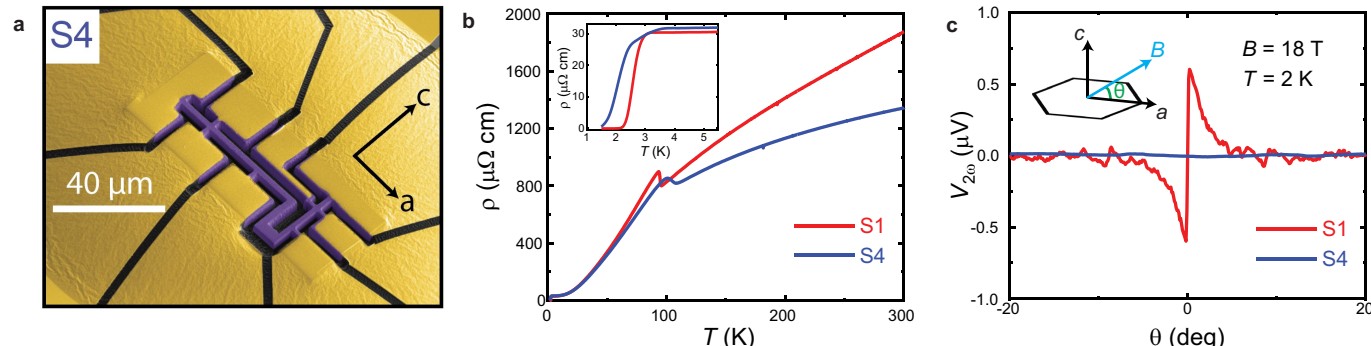

**Extended Data Fig. 7 | Strain effect on eMChA. a**, SEM image of device S4. The sample is attached to the sapphire substrate via a glue droplet. The thin beam along the c-axis allows us to measure the electric response with both current and tensile strain applied along the c-axis. **b**, Temperature dependence of resistivity for S1 and S4. The CDW transition is enhanced to a higher temperature with tensile strain along the c-axis (S4), while $T_c$ is reduced to lower temperature. **c**, Angular spectrum of second harmonic voltage ($V_{2\omega}$). Clearly $V_{2\omega}$ is completely suppressed with tensile strain along c-axis.

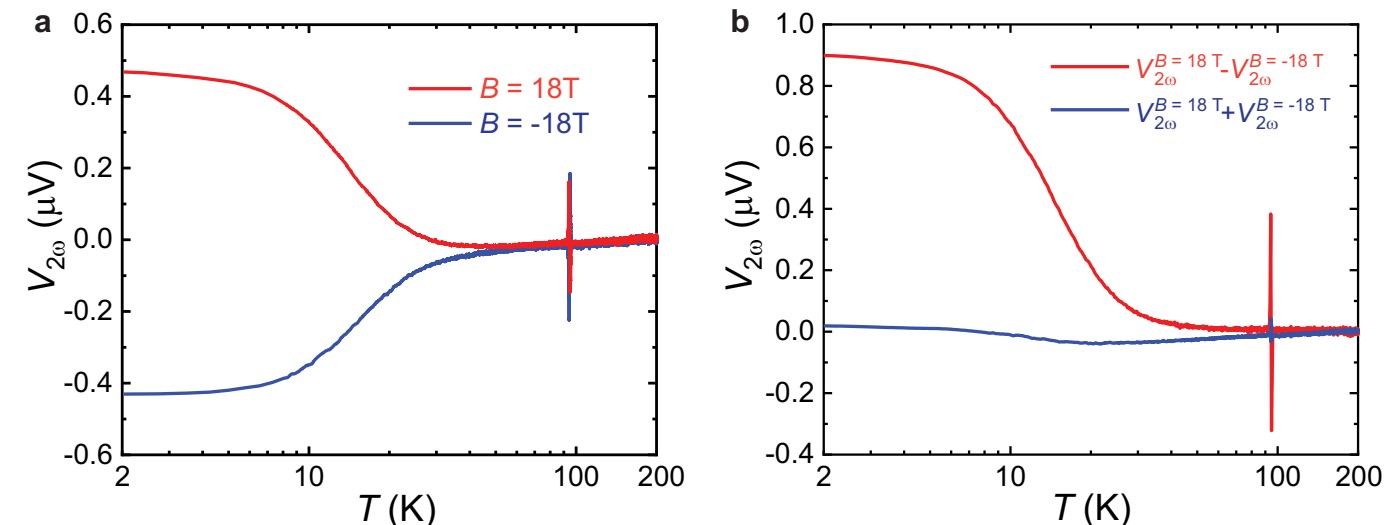

**Extended Data Fig. 8 | Field-symmetry analysis of second harmonic voltage. a**, Temperature dependence of $V_{2\omega}$ measured at $B = \pm 18$ T respectively.
**b**, Temperature-dependent field-symmetric and asymmetric part of $V_{2\omega}$.

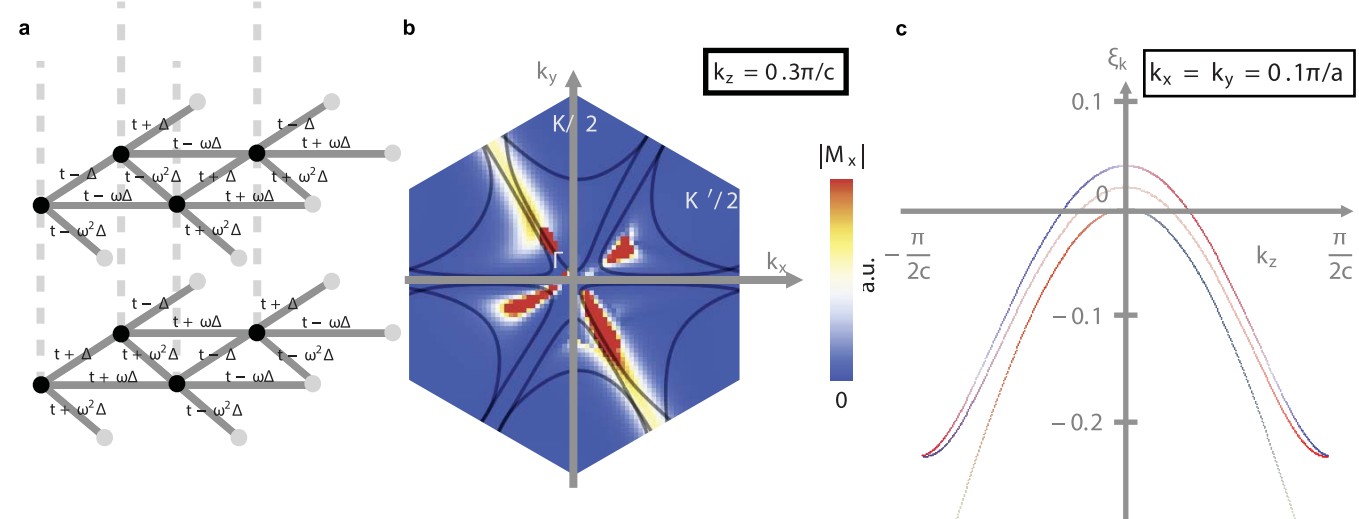

**Extended Data Fig. 9 | Tight-binding model on the stacked triangular lattice with 2 × 2 × 2 chiral CDW order. a**, The unit cell of the model consists of 8 lattice sites (black) on two layers, each harbouring a single electronic orbital. The model is defined by the hoppings as indicated, where $\omega = e^{i2\pi/3}$ (all complex hoppings are oriented to the right). All dashed lines are associated with real hopping amplitude $t_z$. We choose numerical values $t = 1$, $t_z = 0.2$, $\Delta = 0.05$. **b**, Fermi surface of the model for chemical potential $\mu = 2.4$ (black lines)

and orbital magnetic moment component $M_x$ of the states closest to the Fermi level. We observe that $M_x$ is not only finite, but also large for states on the Fermi surface. **c**, Band structure along $k_z$ for a particular choice of $k_x$ and $k_y$, coloured by the orbital magnetic moment component $M_x$, demonstrating that $M_x$ is an odd function of $k_z$ as required for the measured nonlinear response to be nonvanishing.

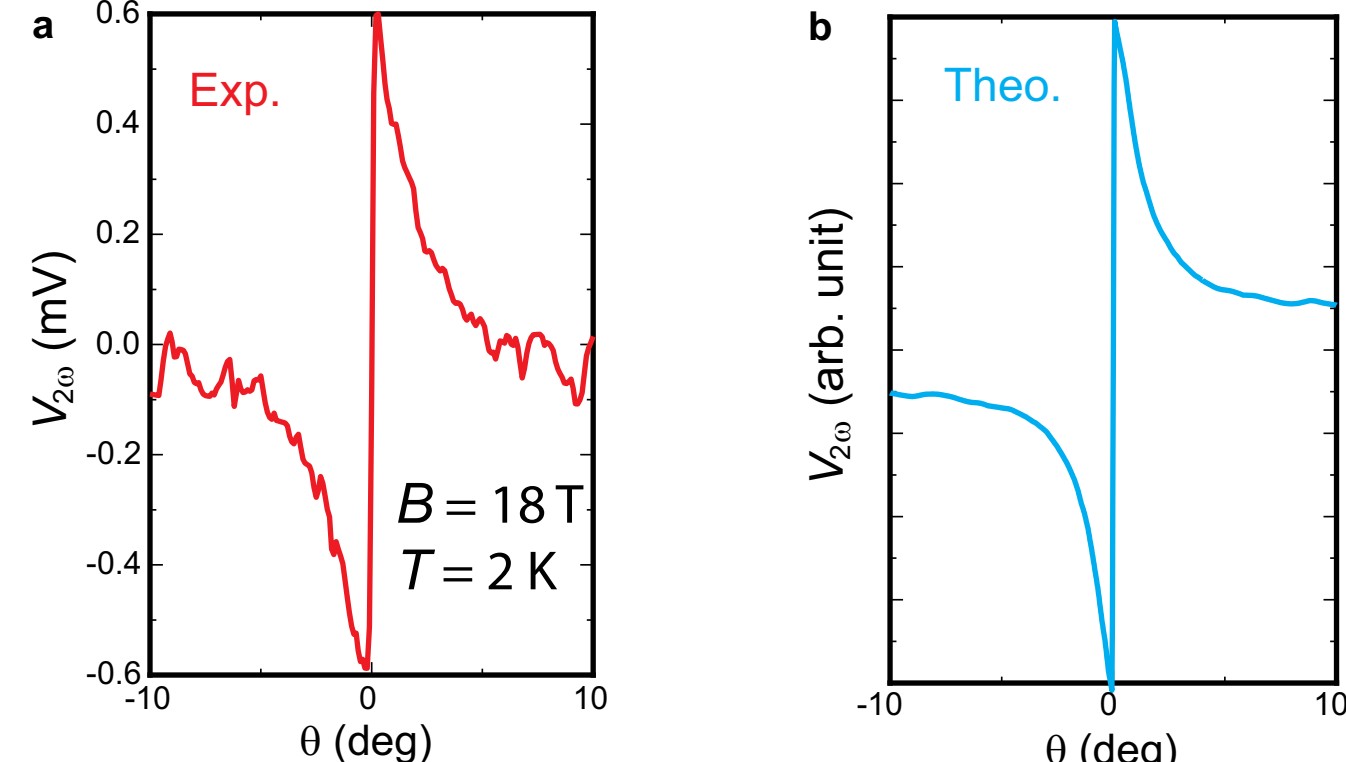

**Extended Data Fig. 10 | Comparison between experimental results and theoretical prediction. a**, Experimentally measured and **b**, theoretically predicted angular dependence of second harmonic voltage $V_{2\omega}$ with $B = 18$ T and $T = 2$ K.