## [Peer Review File · Nature]

Manuscript Title: Switchable chiral transport in charge-ordered Kagome metal CsV₃Sb₅

Redactions – unpublished data

Reviewer Comments & Author Rebuttals

Reviewer Reports on the Initial Version:

Referees' comments:

Referee #1 (Remarks to the Author):

The recently discovered new superconductors AV₃Sb₅ (A=K, Rb, Cs) exhibit several very interesting properties: CDW, unconventional superconductivity, anomalous Hall effect, and possible chiral electronic state due to the time reversal symmetry breaking (TRSB). The last point, namely the TRSB seems very difficult to be proved. The authors here made a microstructure of CsV₃Sb₅ and measured the second harmonic voltage under rotating magnetic field. They observed a large signal of the so-called electronic magneto-chiral anisotropy (eMChA) in the CDW phase of the material, while it becomes significant only at temperatures below $T^* \sim 34$ K, indicative of further evolutions of this ordered phase upon lowering the temperature. They relate this signal to the recently proposed loop-current phase with spontaneously broken mirror symmetries. If this observation is proved to be true and the explanations are more strongly footed, I would be happy to make a positive recommendation. From the data itself, I can read that the $V_{2\omega}$ seems showing the chirality, thus the main observation seems correctly presented. However, since the work may be interesting for special readers, being lacking of general broad interests, thus I would judge that it may be suitable for specialized journals, such as Nature Physics or Nature Nanotechnology. As a specially designed experiment, I have several questions which need to be addressed by the authors, before a recommendation for acceptance is given.

1. Indeed, the $V_{2\omega}$ shows a sharp change when the magnetic field is changed across zero degree (H parallel to the ab-plane), while the real term of magneto resistance reflecting this chirality is $\Delta R/R$. Thus I would like to see a plot, or all plots are drawn in that way. How is the $\Delta R/R$ is changing with the angle, and the field.

2. To prove that the observed signal is really due to the electronic chirality, it is essential to see a current dependence. Is the $V_{2\omega}$ scaled with I^2 , or the $\Delta R/R$ scaled with I . According to formula for eMChA, this seems to be necessary.

3. The relation of $V_{2\omega} \sim B^3$ seems peculiar, thus showing the existence of the chirality. Is there any explanation for this strange behavior.

4. In the loop current model, two components are expected to exist with opposite vorticities of the vortex: one on the hexagons and one on the triangles, they should be opposite in directions. This may be the reason that the residual magnetization is zero at zero field. From the signal observed here, can the authors make an estimate how big the currents or the local fields would be. Or the recent results are not compatible with the proposed model and suggest other models?

Referee #2 (Remarks to the Author):

The authors report a transport signature of electronic magneto-chiral anisotropy in the centro-symmetric layered Kagome metal CsV₃Sb₅. The experimental results are very interesting and may have a high impact on condensed matter physics. To my knowledge, this is the first transport experiment in CsV₃Sb₅ showing macroscopic chiral properties. However, the explanations of the experimental results are very simple (hand-waving type explanation). I cannot recommend to publish this manuscript in Nature in the current form. I have several questions as listed below after I read this paper carefully.

1. Why did the authors use the second harmonic technique for the measurements? How about the first harmonic measurements used for measuring chiral anomaly in weyl semimetals? The authors should provide a detailed analysis in the supplementary information similar to Nature Materials vol 13, 699 (2014). Very detailed derivations and analyses about the second harmonic signals are provided in this NM paper and hence it is quite easy to understand the physics.
2. Could the authors use the detailed derivation of the second harmonic signal to explain the B^3 behaviour in figure 2b? The current explanation is too simple to understand the physics, although the results are very interesting.
3. Could the authors establish a simple model to fit or at least partially fit the interesting data in figure 3? I understand that this is an experimental work. However, even a simple phenomenological model would help readers to have a much deeper understanding of the physics.
4. A fitting or explanation on figure 4a based on the aforementioned simple model should be great.

Referee #3 (Remarks to the Author):

In this manuscript entitled “Field-tuned chiral transport in charge-ordered CsV₃Sb₅”, the authors report non-linear electric response known as electronic magneto-chiral anisotropy (eMChA) in the centro-symmetric layered Kagome metal CsV₃Sb₅, providing another experimental evidence for the spontaneous time-reversal symmetry breaking in the charge-ordered phase of CsV₃Sb₅. Furthermore, the authors demonstrate that the nonlinear magnetotransport can be switched by small out-of-plane fields, pinning down the electronic origin of the time-reversal symmetry breaking. These findings are interesting and may have important applications in chiral electronics. The manuscript is very well written and worth publication. However, the essential physics presented here are known to the community and have already been reported in literature. This work does not improve our understanding of the kagome metals. In this sense, I could not recommend this manuscript for publication on Nature.

Author Rebuttals to Initial Comments:

Dear reviewers,

We thank you for your time and effort to carefully assess our manuscript. We highly appreciate your positive comments about the quality of the work, which truly was a tour-de-force experimental endeavor. At the same time, the review demonstrated the clear need to improve how the importance and reach of the discovery itself are communicated.

We present the first experimental evidence of chiral conduction that can be tuned and even switched by electromagnetic fields. Non-reciprocal transport is a key challenge radiating far outside of our community, with obvious application potentials such as a diode (Wu, H., Wang, Y., Xu, Y. et al. Nature 604, 653–656(2022)). Our work demonstrates that it is possible to parametrically modify, and even invert, the current-voltage characteristic of a diode by magnetic fields in quantum materials.

This is a truly extraordinary materials response, and a first-of-its-kind electronic behavior. Among the few known cases of chiral conductors, the vast majority is structurally chiral (Rikken, G., and Avarvari, N., Phys. Rev. B 99, 245153(2019). Rikken, G., et al., Phys. Rev. Lett. 87, 236602(2001). Pop, F. et al., Nat. Comms. 5, 3757(2014)). Even the enhanced eMChA that emerges in chiral magnets (e. g. Aoki, R. et al., Phys. Rev. Lett. 122, 057206(2019)) ultimately arises from the strong chiral magnetic exchange coupling within structurally strongly chiral crystals.

In retrospect, our previous submission did not emphasize this key point sufficiently. By highlighting the consistency of our results with the known and hypothesized broken symmetries in CsV_3Sb_5 , the manuscript made the observation of chiral transport appear as almost a trivial and natural consequence. Not only did we discover a novel class of switchable diodes, we could further fully rationalize this unconventional behavior. This caused the understandable impression that we did not significantly advance our understanding of CsV_3Sb_5 .

However, it is important to point out how rare efficient chiral conduction in metals is. 65 space groups support chiral crystals, which host many metals. If magnetic space groups and chiral spin textures are included, a vast number of metals are symmetry-

allowed to show chiral conduction. Trivially, any object in a non-mirror-symmetric shape is equally allowed to show chiral conduction. However, the coupling coefficient γ between the dynamics of the itinerant carriers and the chiral potential is usually tiny, which in the vast majority of materials renders the effect undetectable in practice despite its compatibility with symmetry. The fact that CsV_3Sb_5 falls into this rare class of materials with appreciable eMChA is highly non-trivial, especially due to its novel origin from a correlated electronic phase. Our results show this self-consistently, as chiral transport is not observable just below the charge ordering transition, even though presumably there the mirror symmetries are broken. Only at lower temperatures its magnitude is amplified, such that it becomes well observable.

Static chirality has never been tuned by magnetic fields, and hence calls for new explanations. eMChA is a form of non-reciprocal charge transport, which arises from a chirality in the lattice potential such as a chiral CDW. Such chiral charge structures, however, would only couple to magnetic fields at higher order, and clearly not be easily switched by magnetic fields. Here, the strong coupling of different phases is key. Akin to multiferroics, the magnetic field couples to the magnetic structure, possibly a loop current state, which when switched concomitantly switches the chiral charge order.

We hope you enjoy reading the strongly revised manuscript, in which we aim to convey this spirit and main message more clearly. At the same time, we feel it is important to maintain an emphasis on how well this key observation fits into the puzzle of CsV_3Sb_5 . It is fair to say that within complex interacting electron materials, the degree of consistency that emerges even between the early, pioneering experiments truly stands out in a field plagued by sample-to-sample variations and opposing experiments reported by different groups. Despite its complexity, here we may have the rare opportunity to study, and solve, a problem of strongly interacting electrons with multiple interrelated phases.

In the following, we address your technical comments one-by-one.

Referee #1 (Remarks to the Author):

1. Indeed, the $V_{2\omega}$ shows a sharp change when the magnetic field is changed across zero degree (H parallel to the ab-plane), while the real term of magneto resistance reflecting this chirality is $\Delta R/R$. Thus I would like to see a plot, or all plots are drawn in that way. How is the $\Delta R/R$ is changing with the angle, and the field.

Thank you for pointing this out, indeed a discussion of the real term magnetoresistance was missing. Now we present the angular dependence of magnetoresistance and second harmonic voltage at various temperatures. In correspondence to the growth of $V_{2\omega}$, there exists a pronounced peak when a magnetic field is applied within the Kagome plane. Such a magnetoresistance peak, also referred to as coherence peak, is common in the out-of-plane conduction of layered metals (Takatsu, H. et al., Phys. Rev. Lett. 111, 056601(2013). Wu, B. et al., Phys. Rev. Research 2, 022042(2020)). It originates from the open orbit magnetoresistance on a Brillouin-zone-sized quasi-cylindrical Fermi surface. Importantly, $V_{2\omega}$ depends quadratically on magnetoresistance, which is important supporting evidence for the appropriate analysis of chiral conductance in a semimetal with large magnetoresistance as shown later. This is why a plot of $V_{2\omega}$ v. s. ρ_c^2 falls on two straight lines, with the sign of the slope indicating the different chiral states.

Fig. S1 (a) and (b) displays angular dependence of magnetoresistance and second harmonic voltage at various temperatures. (c) summarizes the resistivity dependence of $V_{2\omega}$ at $T = 2$ K and $B = 18$ T.

2. To prove that the observed signal is really due to the electronic chirality, it is essential to see a current dependence. Is the $V_{2\omega}$ scaled with I^2 , or the $\Delta R/R$ scaled with I . According to formula for eMChA, this seems to be necessary.

Fig. S2 (a) and (b) display current dependence of both 1st and 2nd harmonic voltage measured at $B = -18$ T and $T = 2$ K. (c) summarizes the relation between $V_{2\omega}$ and V_{ω} , which shows a parabolic dependence as expected.

Here, we present the current dependence of first and second harmonic voltage signal. Clearly both results show the expected behavior for eMChA. We thank the referee for the comments as this provides further evidence for the second harmonic signal due to electronic chirality. This figure as well as a detailed discussion paragraph have been added to the revised supplementary information.

3.The relation of $V_{2\omega} \sim B^3$ seems peculiar, thus showing the existence of the chirality. Is there any explanation for this strange behavior.

The B^3 -dependence of $V_{2\omega}$ is mainly due to the large linear magnetoresistance of CsV_3Sb_5 . In conventional chiral conductors (such as t-Te), the magnetoresistance is usually very small and can be ignored. In particular, the standard approach of measuring $V_{2\omega}/V_{\omega} \propto \frac{R(B,I) - R(B,-I)}{R(B,I) + R(B,-I)}$ and denoting this as eMChA assumes a constant $R(B,I)$ with B to lowest order, in other words in the denominator. As a consequence, it is customary to only look at the second-harmonic signal $V_{2\omega}$. For a system with large linear magnetoresistance, see Fig. S3 for CsV_3Sb_5 , the resulting form of this eMChA term then seems higher order, namely B^3 . To properly analyze our measurements and show that the response function we effectively observe is indeed lowest order in B and I , we need to analyze the situation for the conductance. Then, as we now discuss in more detail in the supplemental material, Sec. IX, the correct response function is obtained as $\Delta\sigma \approx \frac{V_{2\omega}}{V_{\omega}^2}$. This quantity is indeed linear in both I and B , as we show in Fig.

4 of the main text. This naturally explains the B^3 -dependence of $V_{2\omega}$ displayed in Fig. S3 as $V_{2\omega} \propto \Delta\sigma V_\omega^2$ and both $\Delta\sigma$ and V_ω depend linearly on the magnetic field.

Fig. S3 Field dependence of both c-axis resistivity and 2nd harmonic voltage measured at $T = 2 \text{ K}$.

4. In the loop current model, two components are expected to exist with opposite vorticities of the vortex: one on the hexagons and one on the triangles, they should be opposite in directions. This may be the reason that the residual magnetization is zero at zero field. From the signal observed here, can the authors make an estimate how big the currents or the local fields would be. Or the recent results are not compatible with the proposed model and suggest other models?

We thank the referee for this question. A loop-current model, could indeed result in our observed effect. For instance, the model proposed in Phys. Rev. Lett. 127, 217601(2021) has cancelling loop currents as described by the referee and would be compatible with our observations (specifically the currents are mainly supported on the triangles and oppositely oriented on "up" versus "down" triangles). With this resubmission, we provide a simple toy model (on a triangular lattice such that a single-orbital model is sufficient) to illustrate how an orbital magnetic moment in a band due to mirror symmetry breaking results in a conductance that is first order in both the magnetic field and the electrical current.

However, we caution that our discussion and model calculation should be taken as qualitative only. For a proper and honest estimation of the magnetic moment due to the loop-current and related quantities, detailed knowledge of the band structure, the order parameter, and the coupling to magnetic fields would be required. This is, however, far beyond the scope of our work. The utility of our model calculation is

rather to demonstrate that a system with the assumed pattern of symmetry breaking does indeed support the observed response.

Referee #2 (Remarks to the Author):

1. Why did the authors use the second harmonic technique for the measurements? How about the first harmonic measurements used for measuring chiral anomaly in Weyl semimetals? The authors should provide a detailed analysis in the supplementary information similar to Nature Materials vol 13, 699 (2014). Very detailed derivations and analyses about the second harmonic signals are provided in this NM paper and hence it is quite easy to understand the physics.

We thank the referee for the valuable comments. Indeed the chirality of topological fermions is ideally probed by a first harmonic measurement of the chiral anomaly. However, as it relies strongly on the linear band dispersion of the Weyl pockets, this method is not applicable to topologically-trivial parabolic bands.

In our case, though CsV_3Sb_5 has been theoretically shown to host a topological nodal line, its electronic transport is still dominated by the large Brillouin-zone-sized trivial Fermi surfaces. The non-reciprocal transport signature we study is a chiral magnetotransport feature which depends on the general chirality of the system (structural, magnetic, or electronic) and does not require a linear band dispersion or the notion of conserved chirality within a valley, as for spin-momentum-locked Weyl systems.

On the other hand, the necessity of probing second-order harmonic voltage is due to the general difficulty of capturing the chiral transport signature. As eMChA is a higher-order correction of the transport properties, it is by definition much smaller in amplitude compared to the first-order term. Therefore, it is extremely difficult to probe eMChA via a simple measurement of an IV characteristic. Instead, an AC measurement of all higher-order harmonic voltages allows us to capture the non-reciprocal signature with high accuracy (Fig. S4).

Secondly, we thank the referee for pointing out the reference on a comprehensive analysis of 2nd harmonic voltage due to spin-orbit-torque as AC-current induced an oscillating magnetization and corresponding anomalous Hall effect. Unfortunately, since CsV_3Sb_5 is essentially non-magnetic the analysis of 2nd harmonic voltage induced

by spin-orbit-torque may not be applicable to our results. Instead, the detection of a 2nd harmonic voltage is a direct result of systematic chirality.

Fig.S4 DC and AC IV-characteristics of different harmonic voltages. Due to the smallness of eMChA, it is hard to detect with a standard DC measurement, while easy to capture with the AC measurements of different voltage harmonics.

2. Could the authors use the detailed derivation of the second harmonic signal to explain the B^3 behaviour in figure 2b? The current explanation is too simple to understand the physics, although the results are very interesting.

We thank the referee for pointing out that our discussion was not sufficient, which we fully agree with. We have decided to include a longer discussion of our theoretical considerations in the supplementary material, Sec. IX. For a summary regarding the B^3 -behavior, we refer to our answer to question 3 of Ref. 1.

3. Could the authors establish a simple model to fit or at least partially fit the interesting data in figure 3? I understand that this is an experimental work. However, even a simple phenomenological model would help readers to have a much deeper understanding of the physics.

Indeed, the interested reader will certainly benefit from a simple model to understand our result. We have thus added a paragraph in our paper and added a longer discussion in the supplementary material, Sec. IX.

Crucially, the observed signal needs to be understood as coming from two different angles: First, as we discussed above, in order to understand the B^3 -dependence, the

large linear magnetoresistance needs to be addressed first. While such a magnetoresistance might not be expected in general for a metal, it follows from the density-wave instability and resulting Fermi-surface reconstruction.

Second, for the conductance first order in B and I , the chiral nature of the charge density wave is crucial. Breaking several mirror symmetries, the charge density wave leads to an orbital magnetic moment of the band that couples to the magnetic field. We discuss this phenomenology in the supplementary materials for the case of a toy model with a single orbital. The coupling we discuss leads to the mentioned contribution to the conduction and thus, to the observed second-harmonic signal. Importantly, our model indeed reproduced the experimental results qualitatively (Fig. S5).

Fig.S5 (a) Angular dependence of the first order derivative $\partial(\Delta\sigma)/\partial B$. The green curve represents the model description of chiral conductivity as derived in the supplementary Sec. IX. (b) and (c) represent the experimentally measured and theoretically predicted angular dependence of second harmonic voltage respectively.

4. A fitting or explanation on figure 4a based on the aforementioned simple model should be great.

We thank the referee for this suggestion. To show the qualitative agreement of our simple model with the Results of Fig. 4 in the main text, we have added the prediction of our model, see above.

Referee #3 (Remarks to the Author):

In this manuscript entitled "Field-tuned chiral transport in charge-ordered CsV3Sb5", the authors report non-linear electric response known as electronic magneto-chiral

anisotropy (eMChA) in the centro-symmetric layered Kagome metal CsV₃Sb₅, providing another experimental evidence for the spontaneous time-reversal symmetry breaking in the charge-ordered phase of CsV₃Sb₅. Furthermore, the authors demonstrate that the nonlinear magnetotransport can be switched by small out-of-plane fields, pinning down the electronic origin of the time-reversal symmetry breaking. These findings are interesting and may have important applications in chiral electronics. The manuscript is very well written and worth publication. However, the essential physics presented here are known to the community and have already been reported in literature. This work does not improve our understanding of the kagome metals. In this sense, I could not recommend this manuscript for publication on Nature.

We thank the referee for their expressed interest in our work and the candid comments which resonated well with the other reviewers. The switchability and tunability of chirality is a novel materials response first observed in CsV₃Sb₅ as we argue in the opening statement.

This unusual and surprisingly strong coupling with its tunability will be at the epicentre of future discoveries about the materials chirality. Currently, we work also on the chirality of the superconducting state below $T_C \sim 2.7$ K itself, which similarly shows a striking effect. Close to the upper critical field, inter-vortex repulsion leads to their depinning, and a state characterized by mobile vortices and hence resistance/dissipation. In this liquid state, a pronouncedly chiral response manifests in a strong second-harmonic-generation (Redacted).

While we strongly feel that the discussion of chirality in the flux flow state, with its distinctly different origin, would overload the present scope of the paper, it provides clear evidence that the strong chiral coupling to the electronic system will uncover many unconventional electronic response functions in the future. Most strikingly, here the superconducting state strongly interacts with the underlying chiral texture, providing direct evidence that indeed the different orders in this system are coupled.

Reviewer Reports on the First Revision:

Referees' comments:

Referee #1 (Remarks to the Author):

The authors have addressed my concerns on several issues in the first round review. It is quite hard to detect the possible chiral current state in the AV3Sb5 system, now this experiment presents a clear signal which can be self-consistent explained by the chiral current model. Although it is still not clear how to connect this observation with the local current, I believe this discovery should promote to understand the novel physics in the system. Thus I recomend acceptance for publication.

Referee #2 (Remarks to the Author):

The revised manuscript carefully addressed most of my concerns raised for the previous version.

Chiral orbital current was proposed and investigated in high TC Cuprate and later in iridates and Kagome systems. To date, the material systems with chiral orbital current are still scarce. The authors confirmed a chiral transport behaviour (the gamma BI term) in CSV3Sb5 using the second harmonic technique. All the experiments were well performed. A simple model was employed to explain all the experimental results. This is a pioneering paper using transport methods to investigate chiral orbital transport in Kagome systems. It deserves to be published in Nature Physics or Nature Materials. As a paper published in Nature Materials in 2021 [Nature Materials 20, 1353 (2021)] has already confirmed the chiral charge order using ARPES in a similar Kagome system KV3Sb5, the novelty of this paper is compromised.

I have two more minor questions.

1. A key finding is that the chiral transport in CSV3Sb5 can be tuned by a very small projection of magnetic field to the c direction (Figure 3). It seems that all systems with chiral orbital current should have this property. The authors can discuss this using a few sentences.
2. As long as the resistance changes with the applied current, a second harmonic signal can be detected. The second harmonic signal here comes from the gamma x BI term. Using the first harmonic and the second harmonic, certain parameters can be obtained as shown in page 7. If the authors can do a simple derivation to show the formulae of $V(\omega)$ and $V(2\omega)$ in the supplementary as Professor Kang Wang's Nature Materials (mentioned last time), the paper will be much easier for reading and understanding. I have to derive them by myself when I read page 7.

Referee #3 (Remarks to the Author):

The authors have addressed all questions and comments carefully and have made a big effort to revise the manuscript accordingly. However, I still could not recommend this manuscript for publication on Nature.

The authors present the first experimental evidence of a chiral metal whose chiral transport can be tuned and even switched by external magnetic fields. This kind of material is very rare in nature. The authors did a great job in demonstrating that CsV₃Sb₅ is indeed such kind of chiral metal and the manuscript is very well written. However, given the available experimental facts reported in literature for CsV₃Sb₅, I still would like to view the tunable chiral transport reported in this manuscript as a natural consequence of the spontaneous time-reversal symmetry breaking of electronic origin. Therefore, this work is interesting and worth publication, but not suitable to be published as a Nature paper.

Author Rebuttals to First Revision:

Dear reviewer,

We appreciate your time to carefully assess our revised manuscript. Indeed, we believe that our findings not only add an important aspect to the emergent picture of a highly frustrated, strongly interacting electron system on the Kagome planes of CsV_3Sb_5 , but also open up new avenues to investigate chiral orbital transport using electronic transport methods.

In the following, we address your two technical comments.

1. A key finding is that the chiral transport in CsV_3Sb_5 can be tuned by a very small projection of magnetic field to the c direction (Figure 3). It seems that all systems with chiral orbital current should have this property. The authors can discuss this using a few sentences.

This is an interesting hypothesis, that we now mention in the revised manuscript. In the presence of disorder, this may not be universally the case though. It is well imaginable that the configurational space hosts energy landscapes with deep minima, turning such transitions strongly first order, which again implies the need of a stronger Zeeman field to overcome such barriers. Regardless of such caveats, it is an interesting idea to probe loop current candidate materials with eMChA experiments.

2. As long as the resistance changes with the applied current, a second harmonic signal can be detected. The second harmonic signal here comes from the $\gamma \times BI$ term. Using the first harmonic and the second harmonic, certain parameters can be obtained as shown in page 7. If the authors can do a simple derivation to show the formulae of $V(\omega)$ and $V(2\omega)$ in the supplementary as Professor Kang Wang's Nature Materials (mentioned last time), the paper will be much easier for reading and understanding. I have to derive them by myself when I read page 7.

Thank you for this suggestion. We have added the derivation of V_ω and $V_{2\omega}$ into the supplement, complementing the theoretical derivation of $\Delta\sigma$.